# Query-Aware Graph Attention for Precise Subgraph Retrieval in Knowledge-Augmented Reasoning

## Abstract

In recent years, Retrieval-Augmented Generation (RAG) has demonstrated great potential in enhancing the factual accuracy of large language models (LLMs) in open-domain question answering. Incorporating knowledge graphs (KGs) as external knowledge sources into the RAG paradigm is a promising direction. However, KG-RAG systems for complex multi-hop reasoning tasks still face significant challenges in precisely retrieving structured evidence highly relevant to the query. Existing approaches struggle to dynamically and accurately retrieve graph-based evidence by effectively leveraging query semantics and relational information. To address these challenges, we propose a novel framework called Query-aware Subgraph Retrieval Augmented Generation (QSRAG), centered around a new attention-based architecture termed Query-Relational Graph Attention Network (QR-GAT). QR-GAT is a graph attention mechanism that learns expressive representations of triples by capturing intricate interactions between the query context and relation types. Based on these representations, a scoring module assigns fine-grained relevance scores to triples in the KG, enabling precise subgraph retrieval for downstream reasoning. These structured evidence subgraphs, enriched with confidence scores, are then provided to an LLM to enhance its reasoning capability. Extensive experiments on two widely-used multi-hop Knowledge Graph Question Answering (KGQA) datasets, WebQSP and CWQ, demonstrate that our approach achieves state-of-the-art retrieval performance, particularly excelling in identifying complex multi-hop evidence. KGQA results further show that QSRAG delivers state-of-the-art or competitive performance on both datasets. Our work highlights the effectiveness of query-aware graph attention for accurate structured evidence retrieval, and its potential to enhance knowledge-augmented reasoning with large language models.

## 1 Introduction

The emergence of large language models (LLMs) has significantly advanced the field of natural language processing[Brown et al., 2020, Huang and Chang, 2022, Wei et al., 2022]. However, due to the static and limited nature of their internal knowledge, LLMs often struggle with tasks that demand precise factual grounding or complex reasoning[Kasai et al., 2023, Ji et al., 2023]. Retrieval-Augmented Generation (RAG), which integrates external knowledge into LLMs, has proven to be an effective solution to this limitation[Borgeaud et al., 2022, Gao et al., 2023]. Among external knowledge sources, knowledge graphs (KGs)—structured and large-scale repositories of factual knowledge—are particularly well-suited to support RAG, especially in enhancing LLMs' capabilities in domain-specific or multi-hop question answering[Guo et al., 2024, Pan et al., 2024, Peng et al., 2024].

Submitted to 39th Conference on Neural Information Processing Systems (NeurIPS 2025). Do not distribute.

Effectively integrating knowledge graphs into the RAG paradigm (KG-RAG) requires addressing a key challenge: how to accurately and efficiently retrieve structured evidence (e.g., key triples, paths, or subgraphs) that is highly relevant to a given natural language query from a complex and interconnected graph structure. Although various KG-RAG methods have been proposed, we observe that they generally face three major obstacles:

Firstly, it is still difficult to accurately identify relevant evidence in a multi-hop structure. Answers in knowledge graphs are usually embedded in lengthy multi-hop reasoning paths. Although existing retrieval methods can highlight entities and relationships on the actual reasoning path[Jiang et al., 2023a, Luo et al., 2024b, Sun et al., 2024], they may still be disturbed by irrelevant neighbors or redundant information, resulting in a lack of focus in the retrieval.

Second, insufficient modeling of interactions between query semantics and graph structure hinders performance. Many graph retrieval or GNN-based methods fail to dynamically incorporate the semantic intent of the query into the information aggregation process, instead encoding the graph independently of the query[Mavromatis and Karypis, 2024, Yasunaga et al., 2021, Li et al., 2025]. This limits their ability to adapt attention based on the specific reasoning needs of each question.

Third, retrieval noise and inefficiencies in the usage of LLM contexts present practical challenges. Due to inadequate relevance modeling, retrieved triples often include irrelevant or weakly related facts. Feeding such noisy evidence into LLMs with limited context capacity dilutes useful information[Xu et al., 2023], making it harder for the model to identify and utilize key reasoning evidence. In addition, some KG-RAG methods involve iterative LLM calls during retrieval or reasoning, which substantially increases inference latency[Jin et al., 2024, Gao et al., 2024, Kim et al., 2023, Ma et al., 2024, Xiong et al., 2024, Sun et al., 2024].

Graph neural networks (GNNs) are naturally suited for modeling multi-hop connections in structured data, making them promising tools for subgraph retrieval[Li et al., 2024, Mavromatis and Karypis, 2022]. Their ability to capture complex dependencies among nodes facilitates path discovery and structural reasoning. However, standard GNNs often fall short in query-specific and fine-grained evidence selection. Our work seeks to address this gap by building on the strengths of GNNs while introducing mechanisms for dynamic, query-aware retrieval.

To systematically tackle the above challenges, we propose a novel framework, Query-aware Subgraph Retrieval Augmented Generation (QSRAG). The core of our framework is a query-aware and relation-guided graph attention mechanism called Query-Relational Graph Attention Network (QR-GAT). This mechanism enables efficient and accurate subgraph retrieval from KGs, thereby enhancing downstream LLM reasoning[Brody et al., 2022].

QR-GAT introduces a customized attention mechanism that fuses global query semantics with explicit relation types and applies it to the message passing process of GNNs. This design enables QR-GAT to dynamically adjust the attention to entities and relations based on the input query, thereby solving the problem of insufficient interaction between query semantics and graph structure. By emphasizing entities and relations that are highly relevant to the query, QR-GAT is able to better identify and enhance information on key multi-hop reasoning paths, effectively alleviating the difficulty of evidence alignment in complex graph structures. In addition, with the help of learned node representations and fine-grained attention weights, our computational scoring module is able to accurately score triples based on their relevance to the query. Then, the top-k triples with the highest scores are selected to construct a concise and focused subgraph, which helps reduce redundancy and noise in retrieval results and improves the efficiency and effectiveness of downstream reasoning within the limited context window of LLM.

In the reasoning stage, the retrieved subgraph and its confidence scores are provided to the LLM to guide final answer generation. Notably, unlike some methods [Luo et al., 2024b,a]that require fine-tuning the LLM for specific retrieval formats or tasks, potentially compromising its generality and interpretability, our framework employs LLMs in a plug-and-play manner, leveraging high-quality structured evidence without any fine-tuning. Our key contributions are summarized as follows:

- We propose a knowledge-augmented reasoning framework, QSRAG, to address the challenge of structured evidence retrieval in LLM-based Knowledge Graph Question Answering (KGQA). It is centered on a novel graph attention mechanism, QR-GAT, designed for precise, query-aware, and relation-guided triple scoring and subgraph retrieval.

- We conduct extensive experiments on WebQSP and CWQ datasets, and the results show that QR-GAT achieves state-of-the-art results on standard retrieval metrics.

- We show that integrating subgraphs retrieved by QR-GAT significantly improves LLM reasoning performance, achieving state-of-the-art or competitive KGQA results on both datasets.

- We perform comprehensive ablation studies to validate the role of the query-relation attention mechanism in achieving accurate retrieval, and show that incorporating confidence scores positively impacts LLM reasoning, offering insights into effective evidence utilization.

The remainder of this paper is organized as follows. Section 2 reviews related work in KG-based RAG and graph neural networks. Section 3 introduces preliminaries and problem formulation. Section 4 describes our approach, including the QR-GAT architecture and the two-stage QSRAG framework. Section 5 presents experimental setup and results, followed by further analysis and ablations in Section 6. Section 7 concludes the paper. Additional QA examples and ablation results are provided in the appendix.

## 2 Related Work

**KG-based RAG and Graph Retrieval.** Retrieval-Augmented Generation has become a pivotal paradigm for enhancing the factual capabilities of large language models. Incorporating knowledge graphs as structured external knowledge sources into the RAG framework holds great promise, particularly for complex reasoning tasks. The core challenge in this direction lies in efficiently and accurately retrieving multi-hop structured evidence that is highly relevant to a natural language query from large and complex KGs.To tackle this challenge, various strategies have been proposed. For example, Reasoning on Graphs (RoG) [Luo et al., 2024b] adopts a planning-based approach to retrieve reasoning paths. SubgraphRAG[Li et al., 2025] proposes a trainable subgraph retriever that combines MLP-based scoring with structural features for retrieval. Other methods explore graph retrieval via heuristic search[Sun et al., 2024], combinatorial optimization[He et al., 2024, Hu et al., 2024], or by converting KGs into serialized textual input for LLMs[Wu et al., 2023], as well as biologically inspired memory-augmented retrieval[Gutierrez et al., 2024], non-parametric continual memory frameworks[Gutiérrez et al., 2025], and unified retriever-reasoner architectures for KGQA[Jiang et al., 2023b]. Some approaches also utilize GNNs to assist in graph encoding and retrieval, such as GNN-RAG[Mavromatis and Karypis, 2024], which leverages graph neural representations to support LLM reasoning.While these methods vary in design and purpose, most fall short when it comes to directly assigning fine-grained relevance scores to individual KG triples based on both query semantics and relation types, followed by global ranking for optimal top-k retrieval. For instance, path-based approaches like RoG and subgraph-based scoring methods like SubgraphRAG differ from our attention-based solution in how they model query-relation interactions across multi-hop structures. As a result, the retrieved evidence from these models may be less focused, especially for complex queries.This gap highlights a critical need for more effective mechanisms that can dynamically leverage query and relation information to guide triple-level scoring and filtering on the graph. Our proposed framework directly addresses this need by enhancing the precision of structured evidence retrieval through a more expressive and query-sensitive graph mechanism.

**Graph Neural Networks for KGs.** GNNs are powerful tools for processing graph-structured data, and have been widely and successfully applied to knowledge graphs for tasks such as representation learning, link prediction, and graph-based reasoning. GNNs learn node and relation representations by propagating and aggregating information across graph neighborhoods, effectively capturing structural dependencies. Graph Attention Networks (GATs)[Brody et al., 2022], as a prominent subclass of GNNs, further improve expressiveness by employing attention mechanisms to dynamically weigh neighboring nodes during aggregation. Several KGQA and KG-RAG approaches have leveraged GNNs to handle KG information. For instance, one line of work introduces GRAFT-Net[Sun et al., 2018], which applies a heterogeneous GNN with attention over relations and personalized propagation to perform node classification on question-specific subgraphs that integrate both KG and textual evidence. Other approaches employ GNNs as "subgraph reasoners" to process dense subgraphs retrieved from KGs, or extract shortest paths between entities to construct serialized reasoning chains[Mavromatis and Karypis, 2024]. Variants of GATs[Gao et al., 2024] have been used to explore temporal and structural dependencies within retrieved subgraphs, often combined

with textual representations. While these models demonstrate the utility of GNNs in KG-based tasks, their architectures and applications differ from the objective of fine-grained subgraph retrieval. Most of them operate on pre-filtered or candidate subgraphs, performing downstream tasks such as classification, path extraction, or encoding. Furthermore, their attention and aggregation mechanisms are not explicitly designed to be dynamically modulated by natural language queries and relation types for triple-level relevance scoring across multi-hop structures. The limitations of standard GATs in this context underscore the importance of query- and relation-aware attention for complex retrieval tasks—a capability that has been largely underexplored. Our work builds upon the solid foundation of GNNs and GATs, introducing a novel attention mechanism that explicitly integrates query semantics and relational context into triple-level scoring. The proposed QR-GAT model directly operates on KG structures and dynamically assigns query-conditioned relevance scores to each triple, enabling precise and efficient subgraph retrieval. This offers a new perspective compared to existing GNN-based methods and brings structured evidence retrieval closer to the needs of complex multi-hop KGQA.

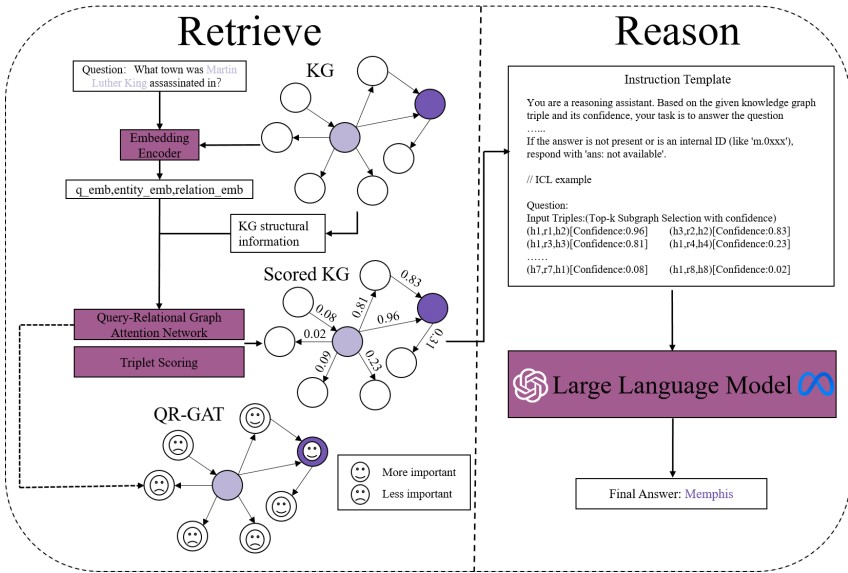

Figure 1: Overview of our QSRAG framework, consisting of (1) a Query-Relational Graph Attention Network (QR-GAT) for structured evidence retrieval from the KG, and (2) a contextual reasoning module using an LLM with in-context learning.

## 3 Preliminaries

In this section, we introduce key concepts relevant to our work, including knowledge graphs, knowledge graph question answering, and retrieval-augmented generation. We also formally define the problem studied in this paper.

**Knowledge Graphs.** A knowledge graph $G$ is a structured representation of factual knowledge in the form of a graph. It typically consists of a set of entities $E$, a set of relation types $R$, and a set of factual triples $T$. Each triple $(h, r, t) \in T$ denotes a fact, where $h \in E$ is the head entity, $r \in R$ is the relation, and $t \in E$ is the tail entity.

**Knowledge Graph Question Answering.** KGQA refers to the task of answering a natural language question $q$ by identifying the correct answer $a$ from a knowledge graph $G$. The answer $a$ may be a single entity or a set of entities in the graph. Solving KGQA typically requires understanding the semantic intent of the question, mapping it to the structure of the KG, and performing reasoning or structured querying to locate the correct answer.

**Retrieval-Augmented Generation.** RAG is a powerful paradigm that combines information retrieval with the generative capabilities of LLMs. In a typical RAG framework, given a user query, a

retrieval module first retrieves relevant knowledge pieces or evidence from an external knowledge source. These retrieved items are then concatenated with the original query and fed into a generation module—typically an LLM—to produce the final response or answer. RAG enables LLMs to incorporate external, real-time, or domain-specific knowledge, thereby reducing hallucinations and improving the factual accuracy and reliability of generation.

**Problem Definition.** We study complex multi-hop KGQA within the RAG framework, where the external knowledge source is a knowledge graph. Formally, given a natural language question $q$ and a knowledge graph $G = (E, R, T)$, the goal is to train a two-stage model to output the correct answer $a$ from $G$. The first stage retrieves a subgraph $S \subseteq T$ containing triples highly relevant to $q$ that support reasoning. In the second stage, the retrieved subgraph $S$ is provided as context to an LLM, which performs multi-hop reasoning via in-context learning (ICL)[Brown et al., 2020] to generate the final answer. These two stages work jointly to achieve accurate, interpretable KG-based QA.

Our main focus is to design an efficient and accurate retrieval module, especially to leverage the query-aware and relation-sensitive graph attention mechanism to accurately extract multi-hop evidence from large-scale knowledge graphs.

# 4 Methodology

We propose a retrieval-augmented generation framework enhanced by knowledge graphs to accurately retrieve structured evidence relevant to a given question and guide LLMs in reasoning. Our framework consists of two main stages: structured evidence retrieval and evidence-based contextual reasoning, as illustrated in Figure 1.

Before building the graph representation, we use the `gte-large-en-v1.5` encoder[Li et al., 2023] to obtain semantic embeddings of entities, relations, and the input question. This yields rich representations: $\mathbf{e}_i$ for entity $v_i$, $\mathbf{r}_{ij}$ for relation $r_{ij}$, and $\mathbf{q}$ for the question.

## 4.1 Structured Evidence Retrieval

The goal of this stage is to extract a subgraph $S \subseteq T$ from the KG that contains triples most relevant to the input question. Rather than using traditional text matching or neighborhood expansion, we model the fine-grained semantic interactions among the query, entities, and relations across multi-hop paths using a novel attention mechanism—**Query-Relational Graph Attention Network (QR-GAT)**.

QR-GAT is designed to guide attention dynamically toward entities and relations that are critical for answering a specific question. It incorporates both query semantics and relation embeddings into the attention computation process. Standard GATs typically treat neighbors uniformly or based only on structure, ignoring query-specific signals. QR-GAT overcomes this limitation by introducing query-and relation-aware attention.

Each node $v_i$ is initialized with:

$$\mathbf{x}_i^{(0)} = \text{Dropout}([\mathbf{e}_i \| \mathbf{q} \| \mathbf{p}_i])$$

where $\mathbf{e}_i \in \mathbb{R}^{d_e}$ is the entity embedding, $\mathbf{q} \in \mathbb{R}^{d_q}$ is the query embedding, and $\mathbf{p}_i$ is a one-hot vector encoding used to label whether the entity is the topic entity.

At each layer $l$, we perform linear projections:

$$\mathbf{z}_i^{(l)} = W_s^{(l)} \cdot \mathbf{h}_i^{(l-1)}, \quad \mathbf{z}_j^{(l)} = W_t^{(l)} \cdot \mathbf{h}_j^{(l-1)}$$

where $W_s^{(l)}$ and $W_t^{(l)}$ are learnable weights for source and target roles respectively.

Attention score $\alpha_{ij}^{(l)}$ is computed by combining structural and query-guided terms:

$$\alpha_{ij,\text{base}}^{(l)} = \mathbf{a}^{(l)\top} \cdot \text{LeakyReLU}(\mathbf{z}_i^{(l)} + \mathbf{z}_j^{(l)} + W_e^{(l)} \cdot \mathbf{r}_{ij})$$

$$\alpha_{ij,\text{plus}}^{(l)} = (W_q^{(l)} \cdot \mathbf{q})^\top \cdot (W_r^{(l)} \cdot \mathbf{r}_{ij})$$

$$\alpha_{ij}^{(l)} = \text{softmax}_j(\alpha_{ij,\text{base}}^{(l)} + \alpha_{ij,\text{plus}}^{(l)})$$

Node representations are updated via multi-head attention:

$$\mathbf{h}_i^{(l)} = \text{LayerNorm}([\mathbf{h}_{i,1}^{(l)}\|\dots\|\mathbf{h}_{i,H}^{(l)}])$$

where $\mathbf{h}_{i,k}^{(l)} = \sum_{j\in\mathcal{N}(i)} \alpha_{ij}^{(l,k)} \cdot \mathbf{z}_j^{(l)}$ for attention head $k$.

We use a bidirectional QR-GAT (BiQR-GAT) to encode both forward and reverse edges. Final entity representation is:

$$\mathbf{h}_i = [\mathbf{h}_i^{\rightarrow}\|\mathbf{h}_i^{\leftarrow}]$$

**Triplet Scoring.** Using final node representations $\mathbf{h}_h$ and $\mathbf{h}_t$ for head and tail entities, the score of a triple $(h, r, t)$ is computed via a two-layer MLP:

$$s(h, r, t) = W_2 \cdot \text{ReLU}(W_1 \cdot [\mathbf{q}\|\mathbf{h}_h\|\mathbf{r}\|\mathbf{h}_t])$$

**Training and Inference.** The retriever is trained as a binary classifier, using a binary cross-entropy loss function to distinguish between positive triplets (triplets on the shortest path between the topic entity and the answer entity) and negative triplets. At inference time, the scores of all candidate triplets are calculated, and the top-$k$ triplets are selected to form a structured evidence subgraph.

### 4.2 Evidence-based Contextual Reasoning

The retrieved top-$k$ triples and their confidence scores are formatted into a textual prompt and fed into the LLM. We adopt an in-context learning (ICL) strategy, where the question and serialized triples are concatenated into a structured prompt of the form: `(head, relation, tail)` `(Confidence:` `0.96)`.

This prompt is then passed into an LLM (e.g., Llama-3.1-8B-Instruct, GPT-4o-mini, or GLM-4-Air), which generates the final answer based on both the question and the supporting evidence. Confidence scores play an important role in guiding the LLM to focus on more relevant triples, improving reasoning reliability and mitigating hallucinations.

To ensure clarity and reproducibility, we provide the full input prompt template as well as several representative multi-hop QA examples in Appendix B. These include both successful and failure cases, which showcase how different triples and confidence patterns affect the final prediction. The prompt examples demonstrate how structured knowledge is effectively integrated into the LLM's context window, and how the LLM reasons over multi-hop paths grounded in the KG.

## 5 Experiments

To thoroughly evaluate the effectiveness of our proposed QSRAG (Query-aware Subgraph Retrieval Augmented Generation) framework, we conduct extensive experiments on two standard Knowledge Graph Question Answering datasets. This section details the experimental setup, evaluation results, and in-depth analysis.

### 5.1 Experimental Setup

**Datasets.** We use two widely adopted KGQA benchmarks: WebQuestionsSP (WebQSP)[Yih et al., 2016] and Complex WebQuestions (CWQ)[Talmor and Berant, 2018], both constructed over Freebase [Bollacker et al., 2008]. WebQSP comprises 4,737 questions requiring up to two-hop reasoning, reflecting relatively simple to moderately complex queries. CWQ includes 34,699 questions with higher compositionality and multi-hop requirements. We followed the preprocessing and data splits of prior work such as RoG.

**Evaluation Metrics.** We evaluate performance at two levels: retrieval and reasoning. For retrieval, triplet Recall@k measures the proportion of retrieved top-k triplets that lie on the shortest path from the topic entity to the answer. Answer Recall@k measures the proportion of answer entities covered by the subgraph formed from the retrieved top-k triplets. For KGQA, we adopt standard evaluation metrics in the KGQA field: Micro F1 measures the overall F1 performance across all question-answer pairs, better reflecting the effectiveness on common questions and questions with many answers;

Macro F1 averages the F1 for each individual question-answer pair, better reflecting the average performance of the method across different question types; Hit evaluates whether at least one of the answers generated by the model is correct; Hit@1 evaluates whether the model's most frequently predicted answer is correct according to any of the ground truth answers.

**Baselines.** We compare QSRAG with several representative KGQA methods covering different technical approaches: SR+NSM w/ E2E[Zhang et al., 2022], a method combining semantic parsing and neural symbolic models, utilizing a pretrained text encoder for constrained path search. Retrieve-Rewrite-Answer[Wu et al., 2023], a multi-stage method typically involving text retrieval, question rewriting, and answer generation. RoG[Luo et al., 2024b] fine-tunes an LLM to predict relation paths and construct reasoning trajectories. G-Retriever[He et al., 2024], combining cosine similarity search with combinatorial optimization techniques, aiming to construct a subgraph capable of connecting topic entities and potential answer entities. GNN-RAG[Mavromatis and Karypis, 2024], utilizing Graph Neural Networks to predict answer entities and extracting triplets on the answer paths through structured methods like shortest path search. SubgraphRAG[Li et al., 2025] is a primary comparison object in this paper, which encodes directional structural distances as structural features using Directional Distance Encoding (DDE) and employs a lightweight multi-layer perceptron (MLP) combined with a parallel triplet scoring mechanism; its retriever design allows for the decomposition of the subgraph distribution.

**Implementation Details.** In terms of hardware environment, we utilize a K100-Ai cluster to complete the entire model training and inference process. For the retrieval stage, in the main experiments, we set the range for the selected number of top-k triplets, k, between 50 and 500, and conduct a detailed analysis of the impact of different k values in subsequent in-depth analysis work. In the reasoning stage, we employ multiple Large Language Models, including Llama-3.1-8B-Instruct, GPT-4o-mini-2024-07-18, and GLM-4-Air-250414. The temperature parameter for these models during inference was uniformly set to 0. In general cases, we primarily used the top 100 triplets for relevant operations, but simultaneously, we also selected representative k values, such as 50 and 200, for study and included the relevant results in the report.

In the research on the retrieval stage, we successfully reproduced the experimental results of SubgraphRAG. After verification, its metrics are consistent with those shown in their paper. Given that this method currently represents a cutting-edge performance level in the field, we directly cited the results from the SubgraphRAG paper for other experiments. In the reasoning stage experiments, we reproduced the experiments of SubgraphRAG, RoG, and GNN-RAG and obtained corresponding results. However, considering that some experiments can no longer represent the most advanced performance level in this field at present, based on strategic considerations, we focused our replication work more on those more updated and more competitive baseline methods.

## 5.2 Evaluation Results

**Retrieval Performance.** Table 1 reports the Triplet Recall@k and Answer Recall@k on WebQSP and CWQ. Our QR-GAT retriever achieves the best results on both datasets. On WebQSP, QR-GAT reaches a Triplet Recall@k of 0.905, improving over SubgraphRAG (0.883) by +2.5%, and over RoG (0.713) by a significant +26.9%. For Answer Recall@k, it achieves 0.942,

Table 1: Retrieval evaluation results on WebQSP and CWQ datasets.Best results are in **bold**.

| | WebQSP | | CWQ | |
|---|---|---|---|---|
| | Triplet Recall | Answer Recall | Triplet Recall | Answer Recall |
| SR+NSM w/ E2E | 0.487 | 0.707 | – | – |
| Retrieve-Rewrite-Answer | 0.058 | 0.740 | – | – |
| RoG | 0.713 | 0.807 | 0.623 | 0.841 |
| G-Retriever | 0.294 | 0.545 | 0.183 | 0.375 |
| GNN-RAG | 0.522 | 0.818 | 0.500 | 0.841 |
| SubgraphRAG | 0.883 | **0.944** | 0.811 | 0.914 |
| **QSRAG** | **0.905** | 0.942 | **0.903** | **0.955** |

nearly matching SubgraphRAG's 0.944 and outperforming all other baselines. On the more challenging CWQ dataset, QR-GAT achieves 0.903 in Triplet Recall@k, surpassing SubgraphRAG (0.811) by +11.3%, and RoG (0.623) by +44.9%. In terms of Answer Recall@k, QR-GAT scores 0.955, improving over SubgraphRAG (0.914) by +4.5% and outperforming all others by a large margin. These results demonstrate the strong capability of our Query-Relational Graph Attention Network (QR-GAT) in accurately identifying and retrieving relevant multi-hop evidence, providing higher-quality structural input for downstream LLM reasoning.

Table 2: KGQA results on WebQSP and CWQ. Best results are in **bold**. By default, our reasoners use the top 100 retrieved triples. Results with 50 and 200 triples (indicated in parentheses) are also shown. Our best result is indicated with an underline.

| | WebQSP | | | | CWQ | | | |
|---|---|---|---|---|---|---|---|---|
| | Micro-F1 | Macro-F1 | Hit | Hit@1 | Micro-F1 | Macro-F1 | Hit | Hit@1 |
| SR+NSM w/ E2E | – | 64.10 | – | – | – | 46.30 | – | – |
| ToG | – | – | 82.60 | – | – | – | **67.60** | – |
| Retrieve-Rewrite-Answer | – | – | 79.36 | – | – | – | – | – |
| G-Retriever | – | 53.41 | 73.46 | – | – | – | – | – |
| RoG | 52.60 | 70.45 | 85.38 | 79.36 | 46.12 | 54.44 | 60.97 | 56.10 |
| GNN-RAG | 10.89 | 71.28 | 85.69 | 80.59 | 28.80 | **59.43** | 66.81 | **61.74** |
| SubgraphRAG + Llama (200) | 42.46 | 66.98 | 83.22 | 77.36 | 39.04 | 41.97 | 54.38 | 48.23 |
| SubgraphRAG + GPT-4o-mini (200) | 49.78 | 69.76 | 85.54 | 78.89 | 44.82 | 43.00 | 53.33 | 48.12 |
| QSRAG + Llama (50) | 47.99 | 69.46 | 83.85 | 79.42 | 46.17 | 45.78 | 56.13 | 51.09 |
| QSRAG + Llama | 48.62 | 70.61 | 85.63 | 80.59 | 43.33 | 45.70 | 56.95 | 51.66 |
| QSRAG + Llama (200) | 49.03 | 70.13 | 85.55 | 79.54 | 40.82 | 44.55 | 57.18 | 51.03 |
| QSRAG + GPT-4o-mini (50) | 51.57 | 69.10 | 81.20 | 75.55 | 49.80 | 44.16 | 51.15 | 46.90 |
| QSRAG + GPT-4o-mini | 55.25 | 70.70 | 83.34 | 76.93 | 50.29 | 45.31 | 53.02 | 48.51 |
| QSRAG + GPT-4o-mini (200) | **55.50** | **71.82** | 84.77 | 78.13 | 49.77 | 45.44 | 54.04 | 48.97 |
| QSRAG + GLM-4-Air (50) | 47.47 | 68.69 | 87.44 | 79.06 | 39.74 | 46.32 | 63.43 | 52.96 |
| QSRAG + GLM-4-Air | 46.38 | 68.25 | 88.85 | 80.59 | 33.06 | 44.16 | 63.15 | 52.54 |
| QSRAG + GLM-4-Air (200) | 45.85 | 68.43 | **90.88** | **81.50** | 29.39 | 44.28 | 65.56 | 53.46 |

**Reasoning Performance.** We systematically evaluate the proposed QSRAG model on two standard question answering datasets, WebQSP and CWQ, and compare it with a series of representative methods (such as SubgraphRAG, RoG, GNN-RAG, etc.). As shown in Table 2, QSRAG performs well on WebQSP. QSRAG + GPT-4o-mini (200) achieves the current best Micro F1 (55.50) and Macro F1 (71.82) scores, which are 5.5% and 1.9% higher than RoG (Micro F1 52.60, Macro F1 70.45), and significantly outperforms SubgraphRAG + GPT-4o-mini (Micro F1 49.78, Macro F1 69.76) by 11.5% and 3.0% respectively. In terms of Hit and Hit@1 indicators, QSRAG + GLM-4-Air (200) achieves the best Hit (90.88) and Hit@1 (81.50) scores. In addition, among different large language models, Hit is always greater than 83 and Hit@1 is always greater than 75, showing consistent and strong recall capabilities.

QSRAG also achieves competitive results on the more challenging CWQ dataset. QSRAG + GPT-4o-mini achieves the best Micro F1 (50.29), 9.0% higher than RoG (46.12) and 12.2% higher than SubgraphRAG + GPT-4o-mini (44.82). GNN-RAG achieves the highest Macro F1 (59.43) among the full entries, while QSRAG + GPT-4o-mini (200) achieves a good Macro F1 (45.44), 5.7% higher than SubgraphRAG + GPT-4o-mini (43.00). In terms of Hit, ToG (67.60) leads most methods, and QSRAG + GLM-4-Air (200) obtains a comparable Hit (65.56). For Hit@1, QSRAG + GPT-4o-mini (200) (48.97) is comparable to RoG (56.10) and leads SubgraphRAG + GPT-4o-mini (48.12) by 1.8%.

On the more challenging CWQ dataset, the SubgraphRAG variants exhibited F1 scores below 45, which may suggest certain limitations in effectively handling complex graph structures compared to their performance on simpler datasets. Overall, QSRAG achieves a good balance between high recall and stable classification performance, showing strong triple reasoning ability and wide applicability.

Table 3: Impact of Retrieved Top-k on Retrieval Performance (Recall@k).

| Topk | WebQSP | | CWQ | |
|---|---|---|---|---|
| | Triple Recall | Answer Recall | Triple Recall | Answer Recall |
| 50 | 0.845 | 0.879 | 0.842 | 0.899 |
| 100 | 0.900 | 0.919 | 0.903 | 0.931 |
| 200 | 0.946 | 0.948 | 0.944 | 0.952 |
| 300 | 0.963 | 0.959 | 0.963 | 0.962 |
| 400 | 0.974 | 0.967 | 0.974 | 0.968 |
| 500 | 0.980 | 0.971 | 0.979 | 0.972 |

# 6 Further Analysis

To gain a deeper understanding of the contributions and robustness of each component of QSRAG, we conducted several analysis experiments.

**Impact of Retrieved Top-k (Retrieval).** Table 3 shows the impact of different numbers of top-k triplets on QSRAG's retrieval performance (Triplet Recall@k and Answer Recall@k) on the WebQSP and CWQ datasets on the test set. The results indicate, as shown in Table 3, that as the number of retrieved top-k triplets increases, the performance of our retriever on the Triplet Recall@k and Answer Recall@k metrics generally shows an upward trend. This suggests that expanding the retrieval scope usually captures more relevant triplets and involved entities. For example, on the WebQSP dataset, as k increases from 50 to 500, Triplet Recall increases from 0.845 to 0.980, and Answer Recall increases from 0.879 to 0.971. A similar trend is observed on the CWQ dataset, where Triplet Recall increases from 0.842 to 0.979, and Answer Recall increases from 0.899 to 0.972. This shows that increasing the number of top-k can provide more potentially relevant evidence.

**Impact of Retrieved Top-k (Reasoning).** Table 4 shows the impact of different numbers of top-k triplets on QSRAG's reasoning performance (Macro-F1 and Hit) on the test set. The results indicate, as shown in Table 4, that QS-RAG's performance generally exhibits a trend of first rising and then falling as the number of top-k triplets increases. This validates that increasing relevant evidence helps improve reasoning accuracy, but there exists an optimal top-k value. Specifically, on the WebQSP dataset, both Macro-F1 and Hit metrics peak at k=100 (Macro-F1 70.61, Hit 85.63), and then gradually

Table 4: Impact of Retrieved Top-k on Reasoning Performance with Llama-3.1-8B-Instruct.

| Topk | WebQSP | | CWQ | |
|---|---|---|---|---|
| | Macro-F1 | Hit | Macro-F1 | Hit |
| 50 | 69.46 | 83.85 | 45.78 | 56.13 |
| 100 | 70.61 | 85.63 | 45.70 | 56.95 |
| 200 | 70.13 | 85.55 | 44.55 | 57.18 |
| 300 | 68.80 | 84.89 | 42.70 | 54.46 |
| 400 | 67.61 | 83.91 | 41.01 | 52.23 |
| 500 | 67.58 | 84.83 | 41.64 | 53.19 |

decrease as the number of top-k increases. On the more challenging CWQ dataset, Macro-F1 achieves its highest score at k=50 (45.78), while Hit peaks at k=200 (57.18), after which the metrics also decline. This phenomenon of performance plateauing or even slightly decreasing after increasing top-k to a certain extent contrasts with the trend of recall continuously increasing with k observed in Table 3. This strongly supports our analysis: although increasing top-k can retrieve more potentially relevant evidence, when the amount of retrieved information exceeds a certain threshold, the introduced noisy triplets may interfere with the LLM's judgment or exceed the LLM's capacity to effectively process the context, thereby impairing the final reasoning performance. This analysis guided our selection of top-k values in the main experiments, where we chose top-k values that performed better on the WebQSP and CWQ datasets as representatives to report.

**Retriever Ablations.** To assess the impact of our query- and relation-aware attention mechanism, we conduct an ablation by replacing the enhanced QR-GAT attention with a standard graph attention (i.e., removing the $\alpha_{ij}^{\text{plus}}$ component and retaining only $\alpha_{ij}^{\text{base}}$). The results are

Table 5: Retriever Ablation Results (Recall@k).

| Model | WebQSP | | CWQ | |
|---|---|---|---|---|
| | Triple Recall | Answer Recall | Triple Recall | Answer Recall |
| Base | 0.851 | 0.900 | 0.877 | 0.940 |
| Plus | 0.905 | 0.942 | 0.903 | 0.955 |

summarized in Table 5. Compared to the full model (`Plus`), removing the query-relation enhancement (`Base`) leads to consistent performance degradation across both datasets. On WebQSP, Triplet Recall drops from 0.905 to 0.851 (-5.97%), and Answer Recall from 0.942 to 0.900 (-4.46%). On CWQ, Triplet Recall decreases from 0.903 to 0.877 (-2.88%), and Answer Recall from 0.955 to 0.940 (-1.57%). This clearly demonstrates the core contribution of our proposed query-aware and relation-guided attention mechanism in accurately identifying question-relevant triples.

# 7 Conclusion

In this work, we address the challenge of accurately retrieving structured evidence for knowledge graph-augmented generation by proposing the QSRAG framework, centered around the Query-Relational Graph Attention Network. QR-GAT enables precise triple scoring and subgraph retrieval through a query-aware and relation-guided attention mechanism. Experimental results demonstrate the strong retrieval performance of our method and its state-of-the-art or competitive KGQA results on both the WebQSP and CWQ datasets. These findings validate the effectiveness of precise structured evidence retrieval—guided by query-aware graph attention—in enhancing knowledge-augmented reasoning.

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

# A  Reasoning Input Ablations.

In addition to simply removing confidence scores, we also explored various strategies to utilize these scores, particularly focusing on filtering the retrieved Top-k triplets based on confidence thresholds. Table 6 presents a comparative analysis of the impact of different confidence filtering thresholds on reasoning performance, measured by Macro-F1 and Hit metrics. The methods compared include using no confidence scores with Llama, using confidence scores with Llama, using confidence scores with GLM-4-Air, and GLM-4-Air at different confidence threshold settings.

The filtering strategy `Confidence > Threshold` retains only triplets with scores above the specified threshold. This approach allows us to assess the effectiveness of confidence-based filtering in enhancing reasoning performance.

Table 6: Reasoning Input Ablation: Impact of Confidence Filtering Thresholds on Reasoning Performance (Macro-F1 and Hit).

| Filter Strategy | WebQSP | | CWQ | |
|---|---|---|---|---|
| | Macro-F1 | Hit | Macro-F1 | Hit |
| No Confidence + Llama | 69.08 | 83.48 | 44.40 | 56.84 |
| Confidence + Llama | **70.61** | 85.63 | **45.70** | 56.95 |
| QSRAG + GLM-4-Air | 68.25 | **88.85** | 44.16 | **63.15** |
| Confidence > 0.0001 + GLM-4-Air | 64.09 | 78.77 | 33.94 | 42.98 |
| Confidence > 0.001 + GLM-4-Air | 64.18 | 78.43 | 35.63 | 45.02 |
| Confidence > 0.01 + GLM-4-Air | 63.37 | 76.17 | 36.77 | 46.58 |
| Confidence > 0.1 + GLM-4-Air | 60.11 | 71.13 | 34.40 | 41.55 |

As shown in Table 6, the method using confidence scores with Llama achieves the best performance on both datasets, with Macro-F1 and Hit metrics higher than those obtained without confidence scores. This suggests that incorporating confidence scores provides valuable signals for fine-grained evidence integration.

When applying threshold filtering with GLM-4-Air, we observe a general degradation in performance as the threshold increases. For instance, on WebQSP, increasing the threshold from 0.0001 to 0.1 results in a decrease in Macro-F1 from 64.09 to 60.11 and in Hit from 78.77 to 71.13. A similar trend is observed on CWQ.

These findings indicate that confidence scores from QR-GAT provide valuable signals to LLMs for fine-grained evidence integration. While threshold filtering may aim to remove noise, it can inadvertently discard useful information or crucial paths. Therefore, feeding all Top-k triplets along with their scores remains the most effective strategy for downstream reasoning.

# B Question-answer Examples.

**System：**
You are a reasoning assistant. Based on the given knowledge graph triple and its confidence, your task is to answer the question.
You must use only the entities found in the triplets that are **meaningful names** (e.g., 'Aviva Stadium', not 'm.0wz2kl3').
Each answer must be a full entity name as it appears in the triplets.
Return each answer in a new line, prefixed with 'ans:'.
If the answer is not present or is an internal ID (like 'm.0xxx'), respond with 'ans: not available'.

**User：**
Input Triplets:
(Lou Seal,sports.mascot.team,San Francisco Giants)[Confidence: 0.9999743700027466]
(San Francisco Giants,sports.sports_team.team_mascot,Lou Seal)[Confidence: 0.9999568462371826]
(2014 World Series,sports.sports_championship_event.champion,San Francisco Giants)[Confidence: 0.8941453695297241]
(2012 World Series,sports.sports_championship_event.champion,San Francisco Giants)[Confidence: 0.8416239619255066]
(2010 World Series,sports.sports_championship_event.champion,San Francisco Giants)[Confidence: 0.8150305151939392]
(San Francisco Giants,sports.sports_team.championships,2014 World Series)[Confidence: 0.7321670055389404]
(San Francisco Giants,sports.sports_team.championships,2012 World Series)[Confidence: 0.6976805329322815]
(San Francisco Giants,sports.sports_team.championships,2010 World Series)[Confidence: 0.6664191484451294]
(San Francisco Giants,time.participant.even2t,2012 World Series)[Confidence: 0.07613193988800049]
(San Francisco Giants,time.participant.event,2010 World Series)[Confidence: 0.07032240927219391]
…
Question: What year did the team with mascot named Lou Seal win the World Series?

**Assistant：**
Based on the given triplets, we can determine that Lou Seal is the mascot of the San Francisco Giants, as indicated by the high-confidence relation (Lou Seal, sports.mascot.team, San Francisco Giants) [Confidence: 0.9999743700027466]. Furthermore, the San Francisco Giants are recorded as champions of the 2014 World Series [Confidence: 0.8941453695297241], the 2012 World Series [Confidence: 0.8416239619255066], and the 2010 World Series [Confidence: 0.8150305151939392]. Therefore, the team with mascot named Lou Seal won the World Series in the years corresponding to those championships.
ans: 2014 World Series
ans: 2012 World Series
ans: 2010 World Series

**User：**
Input Triplets:
(h1,r1,h2)[Confidence:0.96]
(h3,r2,h2)[Confidence:0.83]
…
Question: Which … ?

Please provide your answer in the same format as the example. The answer should be a meaningful entity or information, not a random string like 'm02h98gq' or 'm.0xxx'.Please list all reasonable answers from the above triplets. If the answer is not present in the triplets, respond with 'ans: not available'.

Figure 2: Input prompts for KGQA.

## CWQ-WebQTest-576_01e2da60a2779c4ae4b5d1547499a4f8

**Question:**
Which nation has the Alta Verapaz Department and is in Central America?

**Retrieved Triples:**
Input Triplets:
(Guatemala, location.country.administrative_divisions, Alta Verapaz Department)[Confidence:0.43120062351226807]
(Guatemala, location.location.containedby, Central America)[Confidence: 0.4033266603946686]
(Guatemala, base.aareas.schema.administrative_area.administrative_children, Alta Verapaz Department)[Confidence:0.393011152744]
(Guatemala, common.topic.notable_types, Country)[Confidence: 0.30294546484947205]
(Alta Verapaz Department, location.administrative_division.country, Guatemala)[Confidence: 0.16839228570461273]
(Central America, location.location.contains, Costa Rica)[Confidence: 0.14474257826805115]
(Central America, location.location.contains, Guatemala)[Confidence: 0.13684657216072083]
…
Question: Which nation has the Alta Verapaz Department and is in Central America?

**LLM Response:**
Based on the given triplets, we can determine the following:
1. Guatemala is an administrative division that contains Alta Verapaz Department.
2. Guatemala is also located in Central America.
Therefore, the nation that has the Alta Verapaz Department and is in Central America is Guatemala.The formatted answer is:

ans: Guatemala

**Ground-truth:**
Guatemala

Figure 3: Correct question-answer example 1.

## CWQ-WebQTest-100_524908899a8aa334a18a0ac00f8f2fe6

**Question:**
People from the country that contains Nord-Est Department speak what languages today?

**Retrieved Triples:**
Input Triplets:
(French, language.human_language.countries_spoken_in, Haiti)[Confidence: 0.8332462310791016]
(Haiti, base.aareas.schema.administrative_area.administrative_children, Nord-Est Department)[Confidence: 0.8295478224754333]
(Nord-Est Department, location.location.containedby, Haiti)[Confidence: 0.7992613315582275]
(Haiti, location.country.first_level_divisions, Nord-Est Department)[Confidence: 0.7959402203559875]
(Haitian Creole, language.human_language.countries_spoken_in, Haiti)[Confidence: 0.7403706908226013]
(Haiti, location.country.languages_spoken, Haitian Creole)[Confidence: 0.7223041653633118]
(Nord-Est Department, location.administrative_division.country, Haiti)[Confidence: 0.7090997695922852]
…
Question: People from the country that contains Nord-Est Department speak what languages today?

**LLM Response:**
Based on the given triplets, the country that contains Nord-Est Department is Haiti. The languages spoken in Haiti today are Haitian Creole and French.
Therefore, the formatted answers are:

ans: Haitian Creole
ans: French

**Ground-truth:**
Haitian Creole
French

Figure 4: Correct question-answer example 2.

## WebQSP-WebQTest-1656

**Question:**
What are some of leonardo da vinci most famous inventions?

**Retrieved Triples:**
Input Triplets:
(Aerial Screw, law.invention.inventor, Leonardo da Vinci)[Confidence: 0.9940758943557739]
(Anemometer, law.invention.inventor, Leonardo da Vinci)[Confidence: 0.9930522441864014]
(Diving suit, law.invention.inventor, Leonardo da Vinci)[Confidence: 0.9929106831550598]
(Triple Barrel Canon, law.invention.inventor, Leonardo da Vinci)[Confidence: 0.9921991229057312]
(33-Barreled Organ, law.invention.inventor, Leonardo da Vinci)[Confidence: 0.9916179776191711]
(Double hull, law.invention.inventor, Leonardo da Vinci)[Confidence: 0.9903085827827454]
(Leonardo da Vinci, law.inventor.inventions, Aerial Screw)[Confidence: 0.9774912595748901]
(Leonardo da Vinci, law.inventor.inventions, Anemometer)[Confidence: 0.9757876396179199]
…
Question: What are some of leonardo da vinci most famous inventions?

**LLM Response:**
Based on the given triplets, Leonardo da Vinci is credited with inventing several items. Here are some of his most famous inventions listed in the triplets:

ans: Aerial Screw
ans: Anemometer
ans: Diving suit
ans: Triple Barrel Canon
ans: 33-Barreled Organ
ans: Double hull
ans: Armored car
ans: Parachute
ans: Ornithopter
ans: Viola organista

**Ground-truth:**
Parachute
Diving suit
Aerial Screw
Viola organista
Double hull
33-Barreled Organ
Triple Barrel Canon
Armored car
Ornithopter
Anemometer

Figure 5: Correct question-answer example 3.

**WebQSP-WebQTest-801**

**Question:**
What kind of government is egypt?

**Retrieved Triples:**
Input Triplets:
(Egypt, location.country.form_of_government, Provisional government)[Confidence: 0.9987523555755615]
(Egypt, location.country.form_of_government, Semi-presidential system)[Confidence: 0.9983388185501099]
(Provisional government, government.form_of_government.countries, Egypt)[Confidence: 0.9960112571716309]
(Semi-presidential system, government.form_of_government.countries, Egypt)[Confidence: 0.9928788542747498]
(Egypt, base.aareas.schema.administrative_area.administrative_parent, Earth)[Confidence: 0.00018889721832238138]
(Egypt, location.country.currency_used, Egyptian pound)[Confidence: 3.1110161216929555e-05]
…
Question: What kind of government is egypt?

**LLM Response:**
Based on the given triplets, there are two forms of government associated with Egypt:
1. Provisional government
2. Semi-presidential system
Therefore, the answers are:

ans: Provisional government
ans: Semi-presidential system

**Ground-truth:**
Provisional government
Semi-presidential system

Figure 6: Correct question-answer example 4.

---

**WebQSP-WebQTest-402**

**Question:**
What does joey jordison play in slipknot?

**Retrieved Triples:**
Input Triplets:
(Slipknot, music.musical_group.member, m.010kt51y)[Confidence: 0.6338161826133728]
(Slipknot, music.musical_group.member, m.0h2_x53)[Confidence: 0.28905683755874634]
(Slipknot, music.musical_group.member, m.01ttvxs)[Confidence: 0.12631060183048248]
(Slipknot, music.musical_group.member, m.05nn4sx)[Confidence: 0.12411950528621674]
(Slipknot, music.musical_group.member, m.0h2_x57)[Confidence: 0.10983577370643616]
(m.010kt51y, music.group_membership.group, Slipknot)[Confidence: 0.1037165895104408​3]
(Slipknot, music.musical_group.member, m.05nn4tj)[Confidence: 0.09712643176317215]
…
Question: What does joey jordison play in slipknot?

LLM Response:
Joey Jordison is a member of Slipknot and has played various roles in the band. According to the triplets provided, Joey Jordison has played the following roles in Slipknot:Lead guitar,Backing Vocals,Vocals,Bass guitar.
Therefore, the answers are:

ans: Lead guitar
ans: Backing Vocals
ans: Vocals
ans: Bass guitar

**Ground-truth:**
Drums

Figure 7: Incorrect question-answer example 1.

## CWQ-WebQTrn-1259_1997cb4922db71983be26e6a509950f4

**Question:**
Where did the "Country Nation World Tour" concert artist go to college?

**Retrieved Triples:**
Input Triplets:
(Brad Paisley, people.person.place_of_birth, Glen Dale)[Confidence: 0.9620670676231384]
(Glen Dale, location.location.people_born_here, Brad Paisley)[Confidence: 0.9496924877166748]
(Brad Paisley, music.artist.origin, Glen Dale)[Confidence: 0.6797878742218018]
(Country Nation World Tour, music.concert_tour.artist, Brad Paisley)[Confidence: 0.33858561515808105]
(Colorado State University, common.topic.notable_types, College/University)[Confidence: 0.3122248351573944]
(University of California, Berkeley, common.topic.notable_types, College/University)[Confidence:0.1940333992242813]
…
Question: Where did the "Country Nation World Tour" concert artist go to college?

**LLM Response:**
Based on the given triplets, we need to find the college Brad Paisley attended. The triplets provide information about Brad Paisley's education, including the institution he attended.
From the triplets:
- Brad Paisley, people.person.education, m.0h3d7qj
- m.0h3d7qj, education.education.degree, Bachelor's degree
- m.0h3d7qj, education.education.student, Brad Paisley
The triplet indicates that Brad Paisley received a Bachelor's degree, but it does not specify the institution he attended. The identifier 'm.0h3d7qj' does not correspond to a meaningful name of a college or university in the triplets.
Therefore, the answer is:

ans: not available

**Ground-truth:**
Belmont University

Figure 8: Incorrect question-answer example 2.

