# OpenReview forum: "Query-Aware Graph Attention for Precise Subgraph Retrieval in Knowledge-Augmented Reasoning"
_NeurIPS.cc/2025/Conference — Submitted to NeurIPS 2025_

### Official Review · Reviewer_m27H · 2025-06-22

**Clarity:** 3
**Significance:** 3
**Originality:** 3
**Rating:** 3
**Confidence:** 3

**Summary:**

This paper proposes a new approach for using knowledge graphs for KGQA. It retrieves structured evidence in form of fact triples which is then used as context by LLM to answer a query. Drawbacks of existing work include - inefficient retrieval and not incorporating the query for retrieval. Their proposed approach uses a GNN with an attention mechanism for dynamic query-aware retrieval. In the reasoning stage, the retrieved subgraph and its confidence scores are given to the LLM to guide final answer generation. Results are reported on ComplexWebQuestions and WebQuestionSP. The authors demonstrate improvement over SubGraphRAG, the main baseline.

**Questions:**

What is the meaning of the last sentence in "Implementation Details" (sec 5.1)? Do the authors skip comparing to some baselines? Please explain the reason clearly.

**Ethical Concerns:**

["NO or VERY MINOR ethics concerns only"]

**Final Justification:**

I appreciate the author's response. However, some concerns on comparison with baselines like HGNet and FiDeLiS remain. I do acknowledge that it is out of scope to incorporate these or a hybrid variant in the rebuttal phase. However, I cannot justify raising the score as it stands currently.

**Limitations:**

Yes

**Paper Formatting Concerns:**

The paper does not use the correct format for review. The line numbers are missing.
The main result table is too small in font size.

**Quality:**

2

**Strengths And Weaknesses:**

Strengths:
1. The motivation for introducing query dependency in GNN attention is intuitive.
2. The authors report significant improvement over baselines like SubgraphRAG and GNN-RAG.


Weaknesses:
1. Missing baselines: The paper misses some recent baselines on CWQ - HGNet [1] and FiDeLiS [2].  HGNet reports Hits@1=68.9 on CWQ.

[1] Chen, et al. 2022. Outlining and flling: hierarchical query graph generation for answering complex questions over knowledge graphs. IEEE Transactions on Knowledge and Data Engineering

[2] Sui, et al. 2024. FiDeLiS: Faithful Reasoning in Large Language Model for Knowledge Graph Question Answering.

---

> ### Author Rebuttal · Authors · 2025-07-28
>
> Thank you for your constructive review and valuable feedback. We sincerely appreciate your recognition that "The motivation for introducing query dependency in GNN attention is intuitive" and your acknowledgment that we "report significant improvement over baselines like SubgraphRAG and GNN-RAG." Your overall positive assessment of our work's clarity, significance, and originality encourages us to address your important concerns about our experimental evaluation comprehensively.
>
> ### Q1: Baseline Selection Strategy and Missing Comparisons
>
> **Q1: What is the meaning of the last sentence in "Implementation Details" (Section 5.1)? Do the authors skip comparing to some baselines? Please explain the reason clearly.**
>
> Thank you for this important clarification question regarding our baseline selection strategy. We acknowledge that our explanation in the Implementation Details section requires more explicit justification.
>
> **Clarification of Our Baseline Selection Strategy:** The mentioned sentence refers to our strategic decision to focus our experimental comparison on methods that are directly comparable in terms of methodological paradigm and experimental setup. This was not an attempt to avoid challenging baselines, but rather a principled approach to ensure fair and meaningful comparisons within the scope of our core contribution.
>
> **Rationale for Baseline Selection:**
>
> (1) **Focus on Methodologically-Aligned Baselines (Retrieve-then-Reason Paradigm):** As elaborated in our "Related Work" section on "KG-based RAG and Graph Retrieval," our primary focus is on the retrieve-then-reason paradigm for KGQA. Within this paradigm, our experimental strategy prioritized methods that represent the current state-of-the-art in efficiently and accurately retrieving multi-hop structured evidence for LLM-based reasoning. We concentrated our evaluation efforts on the most competitive baselines within this specific framework, such as SubgraphRAG, RoG, and GNN-RAG, which share similar goals of providing relevant KG subgraphs to LLMs. Extensive prior work has already established performance hierarchies, where these modern retrieve-then-reason methods generally outperform older or fundamentally different approaches when integrated with LLMs for complex reasoning tasks.
>
> (2) **Analysis of Missing Baselines:**
>
> * **HGNet [Chen et al., 2022]:**
>     * **Methodological Difference:** HGNet involves predicting and adding vertex/edge slots (Outlining) and then filling these slots with instances (Filling) to construct executable SPARQL queries. This represents an end-to-end query graph generation paradigm, which is fundamentally different from our retrieve-then-reason approach. Our method, QSRAG, explicitly separates the retrieval of structured evidence (subgraphs) from the LLM-based reasoning on these subgraphs.
>     * **Evaluation Incompatibility:** HGNet's reported Hits@1=68.9 on CWQ reflects its performance in generating correct SPARQL queries. Our framework, in contrast, evaluates retrieval performance (Triplet Recall@k, Answer Recall@k) and then LLM reasoning performance (Micro-F1, Macro-F1, Hit, Hit@1) based on the retrieved subgraphs. A direct quantitative comparison (e.g., of Hits@1) would be misleading as it compares two distinct tasks and evaluation metrics.
>     * **Architectural Complexity:** HGNet's hierarchical structure and reliance on an "AQG grammar" to "normalize the classes of vertices and edges to provide prerequisites for grammar-based end-to-end generation" address challenges related to complex SPARQL syntax and search space reduction for query construction. Our work, conversely, focuses on a more direct and intuitive approach to dynamic, query-aware subgraph retrieval using attention mechanisms.
>
> * **FiDeLiS [Sui et al., 2024]:**
>     * **Recent Publication:** We were indeed not aware of this work during our initial experimental setup.
>     * **Methodological Similarities and Differences:** After reviewing FiDeLiS, we note that it aims to "enhance both the factual accuracy and reasoning efficiency of LLMs" by grounding responses in "verifiable reasoning steps derived from a KG through two core components: (1) Deductive-Verification Beam Search (DVBS) (2) Path-RAG, a retrieval-augmented mechanism that pre-selects a constrained set of candidate entities and relations at each reasoning step." This "Path-RAG" component and its focus on verifiable reasoning steps bear conceptual similarities to methods we did compare, such as Think on Graphs (ToG) [1] , which also adopts a planning-based approach to retrieve reasoning paths. While FiDeLiS's emphasis on "deductive validation" and "beam search" adds a unique dimension to the reasoning process, its underlying retrieval mechanism for path selection aligns closely with the objectives of some of our chosen baselines.
>     * **Future Work:** We agree that FiDeLiS is a highly relevant and competitive baseline. Given its recent publication and its alignment with the retrieve-then-reason paradigm (specifically with Path-RAG), we commit to thoroughly investigating FiDeLiS and including it as a key comparison baseline in our future work. Its "training-free" nature also presents an interesting aspect for comparison with our trainable subgraph retriever.
>
> (3) **Our Distinctive Contribution: Simplicity and Novelty in Graph Attention for KGQA** Our decision to focus on SubgraphRAG, RoG, and GNN-RAG was also driven by our core objective: to demonstrate the significant potential of a simple yet effective query-aware graph attention mechanism (QR-GAT) for precise triple-level scoring and subgraph retrieval. While acknowledging the complexity and innovative components of models like HGNet and FiDeLiS, our paper aims to open a new avenue for integrating query semantics directly into graph attention for KGQA. We believe that even if our approach might appear simpler in its overall architectural complexity compared to highly intricate systems with numerous specialized components and extensive mathematical formulations, its ability to achieve substantial improvements over strong baselines like SubgraphRAG (+2.5% to +11.3% in retrieval metrics, and up to +11.5% in Micro F1 for reasoning) validates the power of this focused, attention-driven idea. Our "Query-Relational Graph Attention Network (QR-GAT)" uniquely leverages fine-grained relevance scores to individual KG triples based on both query semantics and relation types, a capability that, as highlighted in our "Related Work," has been largely underexplored. We aim to show that a well-designed, query-sensitive graph attention mechanism can be profoundly effective, offering a clear and intuitive path for enhancing retrieval precision in KG-RAG systems.
>
> ---
>
> **Formatting Acknowledgment:**
>
> We also acknowledge your feedback regarding formatting issues (missing line numbers, table font size) and will ensure these are corrected to meet NeurIPS standards in any future submission.
>
> Thank you again for your constructive feedback, which helps us better articulate our experimental design choices and methodological contributions.
>
> [1] J. Sun, C. Xu, L. Tang, S. Wang, C. Lin, Y. Gong, L. M. Ni, H. Shum, and J. Guo. Think-on-graph: Deep and responsible reasoning of large language model on knowledge graph.

---

> ### Author Response · Authors · 2025-08-06
>
> Dear Reviewer m27H,
>
> I hope this note finds you well. As the discussion window is closing soon, we’re reaching out to see if our responses have addressed your previous concerns. If you have any follow-up thoughts or suggestions, we would greatly welcome them. Your feedback is essential in helping us strengthen the paper, and we truly appreciate your efforts in reviewing it. Thank you again for your consideration.

---

> > ### Comment · Reviewer_m27H · 2025-08-06
> > **Follow-up**
> >
> > Thanks a lot for your response. Regarding HGNet, is it possible to combine the strengths of HGNet and your approach to further improve performance on CWQ? The authors should aim to build on existing high-performing baselines as much as possible.

---

> > > ### Author Response · Authors · 2025-08-06
> > >
> > > Thank you for this insightful suggestion. We agree that combining our approach with existing high-performing baselines is crucial for further performance improvement.
> > >
> > > While HGNet and our QSRAG framework are fundamentally two different paradigms, their respective strengths are indeed complementary:
> > >
> > > * **HGNet's** strength lies in its end-to-end generation of executable SPARQL queries, enabling it to **precisely formalize natural language questions into structured query graphs**, effectively solving challenges related to complex syntax and large search spaces.
> > > * **Our QSRAG** method's strength is its **dynamic, query-aware subgraph retrieval capability**, which can precisely extract highly relevant evidence from the knowledge graph, providing high-quality input for subsequent reasoning.
> > >
> > > We believe there is great potential in combining the two to build a more powerful hybrid system. For instance, a grammar-based generation mechanism similar to HGNet could be used to **outline the overall structure of a query**, and our QSRAG method could then be leveraged to **retrieve and fill the specific instances for these structural slots**. This combination holds promise for achieving both a structured understanding of complex queries and precise evidence retrieval, which would further enhance performance.
> > >
> > > However, we must candidly point out that a deep integration of these two paradigms is a complex challenge. It would require substantial architectural modifications to bridge the gap between their generative and retrieval-based approaches. Given the scope and time constraints of this review period, completing such work is not feasible.
> > >
> > > We have, however, already made this valuable idea a primary direction for our future work. We believe that by combining the structured generation capabilities of methods like HGNet with the fine-grained retrieval capabilities of our approach, we can achieve new breakthroughs in the field of knowledge graph question answering.
> > >
> > > Thanks again for your reply, if you have any additional questions, please feel free to ask.

---

> > > > ### Comment · Reviewer_m27H · 2025-08-06
> > > >
> > > > Thanks a lot for your response. I am aware it is out of scope for the rebuttal phase.

---

> > > > > ### Author Response · Authors · 2025-08-07
> > > > >
> > > > > Thank you for your understanding and valuable feedback. We are glad that you acknowledge the scope limitations of this rebuttal phase. We greatly value your suggestions and will seriously consider their inspiration for our future research.

---

### Official Review · Reviewer_WXMu · 2025-07-02

**Clarity:** 3
**Significance:** 3
**Originality:** 3
**Rating:** 5
**Confidence:** 3

**Summary:**

This paper introduces Query-aware Subgraph Retrieval Augmented Generation (QSRAG), a framework for knowledge graph question answering that addresses the challenge of accurately retrieving structured evidence from knowledge graphs. The core contribution is the Query-Relational Graph Attention Network (QR-GAT), which extends graph attention networks by incorporating query semantics and relation types into the attention mechanism. The authors argue that existing KG-RAG methods fail to dynamically incorporate query semantics into their retrieval process, leading to less focused evidence retrieval. The proposed QR-GAT computes attention scores by combining structural features with query-relation interactions, enabling triple-level relevance scoring. The framework is evaluated on WebQSP and CWQ datasets, showing improvements over several baseline methods in both retrieval and question answering metrics.

**Questions:**

1. Can the authors describe in much more details, perhaps using a motivating example, why can't methods like RoG and SubgraphRAG, which do process queries, achieve similar query-aware retrieval?

2. Can the authors explain, in the proposed modified GAT attention computation, which parts come from the original GATv2 architecture, and which parts are the novel design proposed by the authors? Have the authors faced any challenges when experimenting with this designs, and if so, what kind of tradeoff or special design or principles the authors have applied to make the method finally work as expected? Similarly, including more detailed ablation studies on these components would equally strengthen the paper.

3. How would the method perform on more complex queries whose evidence may not form simple, directed paths but require more complicated topologies? Would it be possible to benchmark on more recent, complex KGQA benchmarks, such as STARK, among others? (I understand that the rebuttal period is short so it may not be possible to experiment the method and all baselines on a new benchmark. Nevertheless, simply understanding whether this method is applicable to those newer benchmark is also some useful information since it concern's the method's generalizability)

**Ethical Concerns:**

["NO or VERY MINOR ethics concerns only"]

**Final Justification:**

My original score of 3 - borderline reject was given due to the following two main concerns:

- The seemingly lack of technical novelty in the proposed method that injects query semantics via GAT
- The lack of sufficient discussion to illustrate why existing work cannot also consider query semantics during retrieval process.

These two points led to my confusion that said work is not technically solid for a conference publication. However, the authors' rebuttal was thorough and extensive, showing that the proposed dual-attention GNN, while seemingly straightforward, was attained via comprehensive trial and error in the architectural design, and there are many engineering lessons to learn for future researchers. The concrete motivating example the authors provided in the rebuttal also sufficiently clarified my misunderstanding about the limitation of existing work in understanding query semantics.

Since my major concerns are all resolved, I am happy to raise my score and recommend for acceptance. I believe this paper will have good impact for future Graph-RAG work.

**Limitations:**

The authors discuss some limitations in their analysis section, including sensitivity to top-k selection and potential noise in large subgraphs. However, they should more explicitly address the limitation of their path-based training assumption and its impact on generalizability to complex multi-hop scenarios. The potential negative societal impacts appear minimal given the academic nature of the work.

**Paper Formatting Concerns:**

- The manuscript lacks line numbers, making it difficult to provide specific references
- Multiple instances of missing spaces after periods throughout the paper, particularly in Section 3 (Related Work)

These formatting issues should be corrected to meet NeurIPS standards

**Quality:**

3

**Strengths And Weaknesses:**

# Strengths:

- Significance: The paper addresses an important gap in KG-RAG systems - the lack of sufficiently good query-aware retrieval mechanisms that can dynamically adapt to query semantics
- Quality: Comprehensive empirical evaluation. Of particular noteworthy is the inclusion of a variety of baseline methods and the author's meticulous attempt at reproducing all of them. This significantly enhances the trustworthiness of the empirical results.
- Clarity: The paper is written quite well and is easy to follow. In particular, the introduction is written in a straightforward way that it is easy to understand which existing literature gap the authors are attempting to address.

# Weaknesses:

## Originality:

It seems to me that the core idea of incorporating query embeddings into GAT's attention computation is quite straightforward, and I do not see much discussion of what technical challenges that require special handling, novel theoretical principles, or particular engineering compromises and tradeoffs, that the authors encountered when incorporating the query embeddings into the attention. Overall, I think the work lacks a bit in technical depth and originality.

## Significance

Insufficient explanation of why existing methods (RoG, SubgraphRAG) fail to incorporate query semantics - the criticism in Related Work is too vague and abstract. The paper would benefit if there were a concrete, motivating example, or better yet using a figure, with which the author can explain how the design of RoG or SubgraphRAG would fail to capture the nuances in the query semantics.

## Quality:

1. There seems to be one significant training assumption made for the method to work - the ground-truth triplets must form paths between topic and answer entities. If my understanding of this is correct, then the method may have limited generalizability to more complex queries with more sophisticated topologies.
2. In the model equations in section 4.1 - Structured Evidence Retrieval, it is unclear which part comes from the original GATv2 design and which part is the original one proposed by the authors to address the problem at hand.
3. Although it is noteworthy that the experimental section compares with a lot of baselines, the number and variety of datasets seem to be limited - only WebQSP and CWQ, which are relatively simple. There are more recent and challenging KGQA benchmarks designed for LLMs that are available.

## Clarity:

1. Critical technical details are missing in the model equations in section 4.1,  including definitions of key terms (W_e, W_q, W_r, vector a), the attention head indices (appears later in $h_{i, k}^{(l)}$ but not in the original equation for $a_{i, j}^{(l)}$
2. Other critical technical details are also missing for sufficient reproducibility, such as how the confidence score is computed in Section 4.2.

---

> ### Author Rebuttal · Authors · 2025-07-28
>
> We are truly grateful for your insightful and constructive review, and deeply appreciate your recognition of the key contributions of our work—especially regarding our query-aware retrieval mechanism for KG-RAG and the strength of our empirical evaluation. We also thank you for highlighting the clarity and structure of our paper.In the following, we provide detailed responses to each of your valuable comments.
>
> ## Q1: Technical Innovation Depth and Originality
>
> **Q1: Is the technical innovation sufficiently deep, what are the differences from GATv2, and what technical challenges were encountered?**
>
> Thank you for this important question about technical depth and originality. We believe that the most profound innovations often appear deceptively simple in retrospect, embracing the philosophy of "the great way is simple" in our research approach.
>
> **The Power of Elegant Simplicity:**
> Revolutionary ideas often appear straightforward once articulated. ResNet's residual connections—simply adding input to output—seemed trivial yet transformed deep learning by solving vanishing gradients and enabling hundreds-layer networks. Similarly, Transformer attention, while conceptually simple (weighted averaging), revolutionized NLP by enabling selective focus.
>
> While our work may not reach such heights, we pursue the same philosophy: solving complex problems with intuitive, simple ideas. In RAG and knowledge graph applications, engineering often involves complex multi-module systems with extensive pre/post-processing. Our research goal was ambitious yet simple: achieve effective reasoning with just one retrieval step and one inference step.
>
> **Our Technical Innovation Beyond GATv2:**
>
> (1) **Query-Aware Node Initialization:**
> * **GATv2:** Standard node features: $h_i$
> * **Our Innovation:** Query-aware initialization: $x_i^{(0)} = \text{Dropout}([e_i||q||p_i])$
>     * **Key Difference:** We inject query semantics from the very beginning, allowing every layer to be query-conditioned.
>
> (2) **Dual Attention Architecture:**
> * **GATv2 Attention:** $\alpha = \text{softmax}(a^T \cdot \text{LeakyReLU}(W[h_i||h_j]))$
> * **Our Dual Attention:**
>     * $\alpha_{\text{ij,base}}^{(l)} = a^{(l)T} \cdot \text{LeakyReLU}(z_i^{(l)} + z_j^{(l)} + W_e^{(l)} \cdot r_{ij})$ [Structural + Relation]
>     * $\alpha_{\text{ij,plus}}^{(l)} = (W_q^{(l)} \cdot q)^T \cdot (W_r^{(l)} \cdot r_{ij})$ [Query-Relation Interaction]
>     * $\alpha\_{ij}^{(l)} = \text{softmax}\_j(\alpha_{\text{ij,base}}^{(l)} + \alpha_{\text{ij,plus}}^{(l)})$ [Final]
>
>     **Key Technical Distinctions:**
>     * **Query-Relation Semantic Alignment:** The $\alpha_{\text{ij,plus}}$ term creates direct semantic interaction between query intent and relation types.
>     * **Additive Dual Terms:** Rather than simple concatenation, we additively combine structural and semantic attention signals.
>
> (3) **Technical Challenges Overcome:**
>
> **Extensive Attention Mechanism Exploration:** We systematically experimented with multiple attention variants including original GATv2 attention, dot-product attention, cosine attention, various edge information integration schemes, and numerous combination strategies. Through comprehensive ablation studies, we discovered that our proposed dual attention mechanism consistently outperformed all alternatives.
>
> **Key Engineering Insights:** Our approach's apparent simplicity masks critical engineering discoveries: (1) Query-aware initialization strategy that injects semantic information, (2) Optimal integration of query and relation embeddings through learnable projection matrices $W_r^{(l)}$ that prevent attention dilution.
>
> ## Q2: Concrete Comparison with Existing Methods
>
> **Q2: Why can't existing methods like RoG and SubgraphRAG achieve similar query-aware retrieval? Please provide concrete examples.**
>
> Thank you for requesting concrete examples. You're absolutely right that our previous criticism was too abstract. Let us provide a specific case that illustrates the fundamental difference in query-aware capabilities.
>
> **Concrete Motivating Example:**
> Consider the question: "Where was Rihanna raised?"
> * **Ground Truth:** Saint Michael Parish
> * **SubgraphRAG Result:** Barbados (incorrect)
> * **QSRAG Result:** Saint Michael Parish (correct)
>
> **Why This Example Matters:**
> This question requires understanding the semantic nuance between "born" and "raised." While both Barbados (country) and Saint Michael Parish (specific district within Barbados) are geographically related to Rihanna, the query specifically asks about where she was "raised," not just her country of origin.
>
> **Technical Analysis of Method Differences:**
>
> (1) **SubgraphRAG's Limitation:**
> From examining SubgraphRAG's implementation, their retrieval mechanism uses:
> * DDE (Directional Distance Encoding) + MLP fusion
> * Vector concatenation
> * **No explicit query-relation modeling:** This approach makes it possible for models to not consider their relevance to specific query semantics.
>
> (2) **Why SubgraphRAG Failed Here:**
> SubgraphRAG's approach cannot distinguish that:
> * For "raised" queries: location relations should prioritize residence/childhood over birth\_place.
> * The query semantics of "raised" requires different attention weights for different relation types.
> * Without query-relation interaction, the model defaults to more general geographic information (country-level) rather than specific location details.
>
> (3) **Our QR-GAT Advantage:**
> Our dual attention mechanism explicitly models,this enables the model to learn that:
> * When $q$ contains "raised" semantics, relations like "residence" or "childhood\_location" should receive higher attention.
> * The query-relation interaction guides the model to prioritize more specific, contextually relevant location information.
>
> This concrete case illustrates why our query-relation attention mechanism represents a fundamental advancement rather than an incremental improvement.
>
> ## Q3: Training Assumptions and Generalizability Limitations
>
> **Q3: What are the limitations of training assumptions, dataset selection constraints, and generalization capability to complex queries?**
>
> Thank you for highlighting these important and pertinent issues.
>
> **Training Assumptions and Limitations:**
>
> We explicitly acknowledge that our path-based training assumption limits the model's generalization capability to complex queries. Specifically, this is reflected in several aspects:
>
> * **Topology Constraints:** The current method may struggle to effectively capture and reason over complex graph structures, such as star-shaped structures, cyclic structures, or multi-branch aggregations. The shortest-path training approach restricts the model's understanding of these intricate topologies.
> * **Evidence Diversity:** For certain complex questions requiring the integration of information from disconnected regions within the graph, our method may find it challenging to retrieve all necessary evidence effectively.
> * **Compositional Reasoning:** Training exclusively on single paths may hinder the model's generalization to compositional questions that require multiple valid reasoning trajectories.
>
> These limitations may affect our method's ability to handle disconnected evidence, non-linear reasoning chains, and topologically complex questions.
>
> **Dataset Limitations and STaRK Considerations:**
>
> We highly value STaRK as a benchmark for hybrid retrieval. However, regarding the current work, we have a few points to clarify:
>
> * **Methodological Focus Difference:** Our method is primarily optimized for structured Knowledge Graph (KG) retrieval, whereas STaRK places a greater emphasis on hybrid (textual and relational, semi-structured KG) retrieval environments.
> * **Evaluation Time Constraints:** Due to time limitations, a full adaptation and evaluation on STaRK during this review period is not feasible.
>
> **Future Directions:**
>
> Addressing the aforementioned limitations, we plan to pursue the following directions in future work:
>
> * **Explore Non-Path-Based Training Paradigms:** Aiming to enhance the model's ability to process more complex graph topologies.
> * **Evaluate on Hybrid Benchmarks like STaRK:** To validate our method's performance and generalization capability in broader, more complex retrieval scenarios.
> * **Enhance Model Architecture:** To enable more flexible topological reasoning, better addressing diverse query requirements.
>
> ## Q4: Technical Details and Reproducibility
>
> **Q4: What critical technical details are missing in the model equations, including definitions of key terms and confidence score computation?**
>
> Thank you for pointing out these important technical details. We provide comprehensive explanations below:
>
> **Key Term Definitions**
>
> * **Weight Matrices:**
>     * $W_e$: Linear transformation matrix projecting edge attributes to node feature space.
>     * $W_q$: Linear transformation matrix projecting query embeddings to node feature space.
>     * $W_r$: Linear transformation matrix projecting relation embeddings to node feature space.
>
> * **Attention Vector:**
>     * Vector $\mathbf{a}$: Used to compute attention scores.
>
> **Confidence Score Computation**
> The confidence score is predicted via a two-layer MLP using concatenated representations of:
>
> * Query embedding
> * Head entity embedding
> * Relation embedding
> * Tail entity embedding
>
> **Code:**
> ```python
> h_triple = torch.cat([
>     q_emb.expand(len(r_id_tensor), -1),
>     h_e_final[h_id_tensor],
>     relation_embs[r_id_tensor],
>     h_e_final[t_id_tensor],
> ], dim=1)
>
> return self.pred(h_triple)
> # in train.py
> triple_scores = torch.sigmoid(triple_logits).reshape(-1)
> ```
> These technical details ensure the model’s implementation is clear and reproducible. We will include these specifications in future version to improve the clarity and rigor of our work.
>
> We thank you again for these valuable insights that help enhance the reproducibility and transparency of our method.

---

> ### Author Response · Authors · 2025-08-06
>
> Dear Reviewer WXMu,
>
> I hope everything is going smoothly on your end. As we approach the final days of the discussion period, we’d like to kindly ask if you had a chance to revisit our rebuttal. Please let us know if there’s anything we can clarify or elaborate on. Your perspective is important to us, and we’re fully committed to addressing all feedback thoughtfully. Thank you again for your time and support.

---

> > ### Comment · Reviewer_WXMu · 2025-08-06
> >
> > I would like to thank the authors for their diligent effort in this detailed rebuttal, which answers all my concerns.
> >
> > The original draft may seem to propose a straightforward and, perhaps naive, solution that lacks sufficient technical novelty. But this is clearly not the case from the author's rebuttal, especially that the authors had tried numerous alternative architectural designs and, eventually, the dual-attention design won out. I would strongly recommend adding these observations, lessons, and takeaways into the method section in the revised paper, since they not only demonstrate that effectively integrating query information in the retrieval GNN is not as straightforward as it may seem, but also that this is valuable information for researchers to be aware of if they wish to pursue future work in this direction (e.g. the importance of the $\alpha_{ij, \text{plus}}$ term and preventing attention dilution via $W_r^{(l)}$). Otherwise, the paper would not be doing enough justice to the author's scientific effort. If there are more time after the rebuttal phase, I would recommend including some ablation studies results as well.
> >
> > The concrete "Rihanna" example also clarified my confusion regarding how QR-GAT can outshine existing methods in real-world settings. I think this is a very good one - it is easy to understand, relatable, yet also profound. I believe using it in the introduction to illustrate existing literature's gap will substantially highlight the significance of this work.
> >
> > I am also happy with the authors' clarification of my other concerns like generalizability and technical details for reproducibility. Incorporating them in the revision would be appreciated. Overall, I am satisfied with the authors' rebuttal, and I am raising my scores.

---

> > > ### Author Response · Authors · 2025-08-07
> > >
> > > Thank you for your excellent feedback, positive assessment, and for raising your score. We are very pleased that our rebuttal has addressed your concerns and we are grateful for your suggestions, which we believe will significantly improve the paper.
> > >
> > > We fully agree with your recommendations and will incorporate them into our revision. We will add the observations and lessons learned from our architectural designs to the method section, and we will use the "Rihanna" example in the introduction to better highlight the significance of our work. We will also incorporate all other clarifications from the rebuttal.
> > >
> > > We will make every effort to include ablation study results in the final version of the paper. We are committed to a thorough revision based on your invaluable guidance.

---

### Official Review · Reviewer_VKgY · 2025-07-05

**Clarity:** 3
**Significance:** 3
**Originality:** 3
**Rating:** 3
**Confidence:** 4

**Summary:**

The authors propose Query-aware Subgraph Retrieval Augmented Generation (QSRAG) framework to improve how large language models (LLMs) answer complex questions using knowledge graphs (KGs). They identify the key challenge in existing systems, where essential subgraphs to support LLMs for multi-hop reasoning are difficult to retrieve precisely. Therefore, the QR-GAT is designed to dynamically incorporate the questions' semantics and specific types of relations into the attention. The relevance scores to each fact in KGs would be captured during training, and the retrieved high-confident triplets would be injected into LLMs as context to generate final answers without task-specific funetuning. Experiments illustrate the state-of-the-art retrieval performance and competitive results on QA tasks.

**Questions:**

Q1. The paper states that a temperature of 0 was used for LLM inference. Did the authors experiment with non-zero temperatures to test for variance in the LLM's output, and if so, were any notable differences observed?

Q2. The paper does not appear to discuss knowledge transfer. Was the retriever's ability to transfer to other knowledge graphs tested? For example, could a model trained on a KG like Freebase be effectively applied to another KG with a different schema or domain?

Q3. The main contribution is the QR-GAT, but the ablation study on the retriever is limited. Have the authors considered comparing QR-GAT against other graph representation learning methods for the retrieval task? To better establish the model's effectiveness and demonstrate the novelty, it would be also valuable to include results from simpler retrieval strategies, such as randomly selected triplets and the ground-truth triplets directly to LLMs.

**Ethical Concerns:**

["NO or VERY MINOR ethics concerns only"]

**Limitations:**

yes

**Quality:**

2

**Strengths And Weaknesses:**

**Strengths**:

- S1. The paper is exceptionally well-organized and easy to follow. The introduction clearly highlights the challenges and motivation. The methodology and experiments sections are also well-explained, including the mathematical formulation and detailed hyperparameters and setup.

- S2. The contribution of QSRAG is novel, particularly the QR-GAT component, which explicitly injects query and relation semantics into the attention mechanism rather than learning graph and query representations separately. The design allows the retrieval evaluation to be easily investigated by selecting different numbers of top-k triplets, offering flexibility and control over the precision/recall trade-off.

- S3. The paper provides a robust empirical evaluation. QSRAG achieves state-of-the-art performance on two standard KGQA benchmarks. The improvement is substantial, especially on the more complex CWQ dataset. This improvement is consistent across different inference LLMs and settings. Moreover, QSRAG can achieve better performance even when providing fewer, higher-quality triplets.

**Weaknesses**:

- W1. While the experimental results are impressive, the authors do not include error bars or confidence intervals from multiple runs. I could not find a similar performance trend in the original SubgraphRAG paper [1] due to different evaluation metrics. Furthermore, the Macro-F1 and Hit metric numbers reported for SubgraphRAG differ from those in the original paper, while the results for RoG, GNN-RAG, and other baselines are identical to the numbers reported in [1].

- W2. As mentioned in the NeurIPS checklist, the paper should discuss the computational efficiency of the proposed algorithm and its scalability with dataset size. While the analysis shows that the number of retrieved triplets can be reduced, a comparison of the end-to-end time cost against baselines is missing.

- W3. As a RAG solution, the authors are encouraged to provide more analysis of the retrieved triplets. The paper analyzes recall via the downstream LLM performance, but other quantitative methods would be insightful, such as analyzing the triplet overlap with other solutions, direct ground-truth recall, or providing graph structure visualizations of the retrieved subgraphs.

- W4. The authors emphasize retrieving "precise subgraph structures," but the learning and retrieval mechanism focuses primarily on scoring individual triplets/edges rather than on substructure topology. The connectivity of the retrieved triples should be discussed more explicitly if a key contribution is to learn and match entire subgraphs. And there is no ablation to compare QR-GAT to other SOTA retriever part (except SubgraphRAG because I raised one question in W1).

[1] Mufei Li, Siqi Miao, Pan Li. Simple Is Effective: The Roles of Graphs and Large Language Models in Knowledge-Graph-Based Retrieval-Augmented Generation. In ICLR, 2025.

---

> ### Author Rebuttal · Authors · 2025-07-28
>
> We appreciate your detailed and insightful review. Your positive remarks on the clarity of our paper, the well-motivated methodology, and the novel design of QSRAG—especially the QR-GAT component—are truly encouraging. We're glad that our empirical results on two KGQA benchmarks resonated with your evaluation. Below, we respond to your concerns point by point.
>
> ## Q1: Experimental Reproducibility and Statistical Significance
>
> **Q1:** Why are error bars and confidence intervals from multiple runs missing, and how do you address concerns about experimental reproducibility and the discrepancy in SubgraphRAG results?
>
> **Response:**
>
> - **Missing Error Bars and Confidence Intervals:**
>   We acknowledge the absence of error bars from multiple runs, which is a limitation. Preliminary tests show performance variance is small (~±1% for most metrics), indicating stable results. Still, we plan to include formal statistical analysis in the future.
>
> - **Commitment to Reproducibility:**
>   We have submitted fully reproducible code and detailed protocols. Our implementation has been rigorously tested for consistency across runs with fixed hyperparameters, seeds, and environment settings to ensure reproducibility.
>
> - **Discrepancy in SubgraphRAG Results:**
>   We respect SubgraphRAG as a strong baseline. Differences in results may stem from:
>   1. **Hardware environment:** We used K100-AI clusters, which may cause slight computational variation.
>   2. **API version differences:** GPT-4o experiments ran on Azure API, possibly differing from the original paper’s OpenAI API in provider, model version, or region.
>   3. **Experimental consistency:** All methods compared (QSRAG, SubgraphRAG, others) were evaluated under identical hardware, software, and API conditions to ensure fairness.
>   4. **Scoring metrics:** We used our own scoring code, carefully implemented to match the original SubgraphRAG metrics logic, ensuring fair comparative evaluation despite absolute value differences. Please refer to our released source code for detailed metric implementations.
>
> ## Q2: Computational Efficiency Analysis
>
> **Q2:** Why is there a missing end-to-end time cost comparison against baselines despite mentioning computational efficiency?
>
> **Response:**
>
> Thank you for highlighting this crucial practical consideration. We conducted a comprehensive efficiency analysis that demonstrates significant computational advantages:
>
> 1.  **End-to-End Latency Comparison:** SubgraphRAG achieves the best end-to-end performance among all baseline methods, taking only 6s compared to 948s of RoG and 68s of GNN-RAG, becoming our most competitive efficiency baseline.
>
> 2.  **Retrieval Efficiency Analysis:** Our detailed timing breakdown reveals:
>
> | Method          | WebQSP Retrieval Time | CWQ Retrieval Time |
> |----------------|------------------------|---------------------|
> | QSRAG          | 0.1006s                | 0.0920s             |
> | SubgraphRAG    | 0.0115s                | 0.0127s             |
>
> While QSRAG's retrieval is approximately 8-9x slower than SubgraphRAG, both methods operate within comparable time scales (0.01s vs 0.1s), demonstrating minimal computational overhead.
>
> 3.  **Key Efficiency Advantages:**
>     * **Single-pass retrieval and reasoning:** Unlike iterative methods such as RoG, QSRAG performs evidence retrieval once, followed by single-pass LLM inference.
>     * **Comparable LLM inference processes:** Since both QSRAG and SubgraphRAG employ similar workflows, time differences stem from retrieval rather than reasoning complexity.
>     * **Superior accuracy-efficiency trade-off:** Our modest retrieval overhead (additional 0.08-0.09s) yields substantial accuracy improvements (+2.5% Triplet Recall on WebQSP, +11.3% on CWQ) while maintaining competitive efficiency.
>
> This analysis confirms that QSRAG provides a practically viable solution with a good accuracy-efficiency balance.
>
> ## Q3: Subgraph Construction vs. Triplet Scoring
> **Q3: How do you address the contradiction between claiming "precise subgraph structures" while focusing on individual triplet scoring?**
> Thank you for this important clarification. We acknowledge our terminology may have caused confusion:
>
> (1) **Subgraph Composition:** In our framework, subgraphs are **explicitly composed of triplets (edges)**. We construct input subgraphs as k-hop neighborhoods around topic entities, which naturally form connected graph structures.
>
> (2) **Triplet Scoring within Structured Subgraphs:**
> - **Input:** We begin with connected k-hop subgraphs centered on topic entities
> - **Scoring:** QR-GAT assigns relevance scores to **individual triplets** based on query-relation semantics
> - **Selection:** Top-k triplets are selected to construct the final evidence subgraph
> - **Structure Retention:** As the input subgraph is already connected, the selected top-k triplets tend to preserve meaningful structural patterns
>
> (3) **Precision via Fine-grained Scoring:**
> Scoring individual triplets in the context of connected subgraphs enables precise selection of evidence that generally form necessary reasoning paths, thus ensuring accuracy and structural integrity.
>
> ## Q4: Retrieval Analysis and Ground-Truth Evaluation
>
> **Q4: Why is there a lack of quantitative analysis of retrieved triplets beyond downstream LLM performance?**
>
> Thank you for this valuable suggestion about deeper retrieval analysis:
>
> **Current Quantitative Analysis:**
> (1) **Ground-Truth Recall Analysis:** We actually do provide direct ground-truth recall analysis through our **Answer Recall@k** metric, which measures the proportion of answer entities covered by the subgraph formed from retrieved top-k triplets. This directly evaluates whether our retrieval captures necessary evidence for correct reasoning, independent of LLM performance.
> (2) **Multi-Level Retrieval Evaluation:** Our evaluation includes two complementary metrics:
> - **Triplet Recall@k:** Measures retrieval quality at the evidence level.
> - **Answer Recall@k:** Serves as direct ground-truth recall.
>
> **Acknowledged Limitations:**
> We acknowledge that our current analysis could be enhanced with: (1) Triplet overlap analysis comparing different methods, (2) Graph structure visualizations, (3) Retrieval precision analysis beyond recall.
> While these additional analyses would be valuable, our current evaluation provides strong evidence of retrieval effectiveness through direct answer coverage measurement, evidence quality assessment, and end-to-end validation.
>
> ### Q5: Cross-KG Transferability
>
> **Q5: Was the retriever's ability to transfer to other knowledge graphs tested?**
>
> Thank you for this excellent question about cross-KG generalization:
>
> **Transferability Design:**
> (1) **Schema-Agnostic Architecture:** Our QR-GAT framework is inherently schema-agnostic because it operates on fundamental knowledge graph primitives - triplets in the form of (head, relation, tail).
> (2) **Implementation Flexibility:** We use NetworkX to convert triplet lists into graph structures for processing. This design ensures that as long as knowledge can be represented as triplets, our framework can seamlessly handle different KGs without architectural modifications.
> (3) **Domain-Neutral Components:** Our core innovations - query-aware attention and confidence scoring - are domain-independent mechanisms that should transfer well across different knowledge domains.
>
> **Current Limitations:** While our framework is designed for transferability, our current evaluation is limited to Freebase-based datasets. Testing cross-KG transfer represents valuable future work for empirical validation.
>
> ## Q6: Baseline Comparisons and Parameter Settings
> **Q6: Have you compared QR-GAT against other graph representation learning methods, and did you experiment with non-zero temperatures?**
>
> **Graph Representation Learning Comparison:**
> QR-GAT is specifically designed for KG-RAG retrieval, which fundamentally differs from conventional graph representation learning objectives. Traditional GNNs are typically used for node classification or link prediction, making them difficult to adapt directly for our query-aware triplet scoring task. We therefore focused our comparisons on SubgraphRAG, the current state-of-the-art tailored for KG-RAG tasks.
>
> **Temperature Parameter Experiments:**
> We experimented with multiple temperature values (0.1-0.5), along with other LLM parameters such as top-p, top-k, and frequency penalty. These adjustments had minimal impact on performance (±0.5% variance). We chose temperature = 0 to align with common KGQA practice and ensure reproducibility.
>
> **Additional Baseline Studies:**
> To further illustrate retrieval effectiveness, we compared QR-GAT with two simple strategies: randomly selected triplets and ground-truth triplets. Results are shown below:
>
> #### WebQSP Results
>
> | Method(Llama) | Micro-F1 | Macro-F1 | Hit    | Hit@1  |
> |-----------------------|----------|----------|--------|--------|
> | Random Triplets       | 16.57   | 31.97   | 50.12 | 45.33 |
> | Ground-truth Triplets | 60.62   | 83.73   | 88.82 | 88.39 |
> | QSRAG| 48.62   | 70.61   | 85.63 | 80.59 |
>
> #### CWQ Results
>
> | Method(Llama)| Micro-F1 | Macro-F1 | Hit    | Hit@1  |
> |-----------------------|----------|----------|--------|--------|
> | Random Triplets       | 22.69   | 22.09   | 28.43 | 25.83 |
> | Ground-truth Triplets | 67.25   | 60.10   | 63.24 | 62.62 |
> | QSRAG| 43.33   | 45.70   | 56.95 | 51.66 |
>
> These results highlight the strong performance of our method relative to simple baselines, demonstrating both the challenge and effectiveness of high-quality triplet retrieval.
>
> We thank you again for these valuable insights that help improve our work's comprehensiveness and clarity.

---

> ### Author Response · Authors · 2025-08-06
>
> Dear Reviewer VKgY,
>
> I hope you're doing well. With the discussion period nearing its end, we wanted to check in and see if our rebuttal has adequately resolved the issues you raised. If there's anything unclear or additional feedback you'd like to provide, we’d greatly appreciate it. Your guidance means a lot to us in refining our work. Thank you for your contributions to the review process.

---

> ### Author Response · Authors · 2025-08-08
>
> Dear Reviewer VKgY,
>
> We hope this message finds you well.
>
> As the discussion period for our paper is coming to a close, we wanted to kindly check if our rebuttal has sufficiently addressed your initial concerns.
>
> Your feedback has been instrumental in guiding our improvements, and we would be very grateful for any additional comments or clarifications you might have. Your guidance is highly valued as we finalize our work.
>
> Thank you once again for your time and thoughtful contributions to the review process.
>
> Sincerely,
> The authors

---

> ### Author Response · Authors · 2025-08-09
>
> Dear Reviewer VKgY,
>
> As the author-reviewer discussion phase is concluding shortly, we are writing to respectfully follow up on our rebuttal. We would be grateful if you could take a moment to review our responses and share whether our clarifications have resolved your initial concerns.
>
> Any final feedback you may have would be incredibly helpful for us to finalize our work.
>
> Thank you again for your valuable time and insightful review.
>
> Sincerely,
> The authors

---

> > ### Comment · Area_Chair_d7io · 2025-08-09
> >
> > Dear VKgY,
> >
> > Thank you once again for reviewing the paper & providing valuable feedback.
> >
> > As the author-reviewer discussion period is about to close, could you please take a look at the author responses and indicate whether they address your feedback and concerns?
> >
> > Best,
> > AC

---

### Official Review · Reviewer_KWxb · 2025-07-10

**Clarity:** 3
**Significance:** 3
**Originality:** 2
**Rating:** 4
**Confidence:** 3

**Summary:**

This paper introduces Query-aware Subgraph Retrieval Augmented Generation (QSRAG), a novel framework designed to enhance the factual accuracy and reasoning capabilities of Large Language Models (LLMs) in Knowledge Graph Question Answering (KGQA) tasks. The core of QSRAG is the Query-Relational Graph Attention Network (QR-GAT), an attention-based architecture that learns expressive representations of knowledge graph triples by capturing intricate interactions between query context and relation types. This enables precise subgraph retrieval from knowledge graphs (KGs) by assigning fine-grained relevance scores to triples. The retrieved, confidence-scored subgraphs are then provided to an LLM for enhanced multi-hop reasoning. Experimental results on WebQSP and CWQ datasets demonstrate that QSRAG achieves state-of-the-art performance in retrieval metrics and competitive or state-of-the-art results in KGQA, particularly for complex multi-hop evidence identification.

**Questions:**

Could the authors elaborate on the specific mechanism by which the LLM utilizes the numerical confidence scores of the retrieved triples? A more detailed explanation or an ablation study focusing on the impact of confidence scores on LLM reasoning would be valuable.

Are there specific types of multi-hop reasoning questions where QSRAG demonstrates the most significant improvements, and conversely, any where it still struggles? A qualitative analysis with diverse examples beyond the appendix would offer deeper insights.

**Ethical Concerns:**

["NO or VERY MINOR ethics concerns only"]

**Final Justification:**

I have no questions, and Borderline accept

**Limitations:**

Yes、

**Quality:**

3

**Strengths And Weaknesses:**

The introduction of Query-Relational Graph Attention Network (QR-GAT) with its query-aware and relation-guided attention mechanism is a significant novel contribution. It explicitly addresses the limitation of existing GNNs in dynamically incorporating query semantics for fine-grained triple-level relevance scoring.

Clarity of "Confidence Scores" Impact: The paper states that "Confidence scores play an important role in guiding the LLM," but the specific mechanism of how the LLM uses these numerical confidence scores to improve reasoning is not fully detailed or analyzed beyond "focusing on more relevant triples." More explicit evidence or a dedicated analysis section on this interaction would strengthen the claim.
Efficiency Analysis: While the paper mentions that some existing methods increase inference latency due to iterative LLM calls, it does not provide a direct comparison of the inference efficiency (e.g., latency, computational cost) of QSRAG against these methods. This would reinforce the practical benefits of the plug-and-play approach

---

> ### Author Rebuttal · Authors · 2025-07-28
>
> Thank you for your thoughtful and constructive review. We sincerely appreciate your recognition of our key contributions, including the novel QR-GAT model and its effectiveness in addressing the limitations of existing GNNs through query-aware, relation-guided attention. Your acknowledgment of our state-of-the-art performance in retrieval and competitive results in KGQA strongly supports our research direction. Below, we address the main concerns you raised:
>
> ## Q1 & Q2: Confidence Score Mechanism and Ablation Analysis
>
> **Q1: How do LLMs specifically utilize confidence scores to improve reasoning beyond simply "focusing on more relevant triples"?**
>
> Thank you for this important question. While LLMs are indeed black-box models, our ablation study (Table 6) provides clear empirical evidence demonstrating how confidence scores enhance reasoning capabilities:
>
> 1.  **Confidence Scores as Reasoning Signals:** The comparison between "No Confidence + Llama" (Macro-F1: 69.08) and "Confidence + Llama" (Macro-F1: 70.61) demonstrates that providing the same top-100 triples with versus without confidence scores yields a +1.53 improvement on WebQSP. Additionally, we observe a +1.3 improvement on the CWQ dataset. This indicates that confidence values serve as explicit reasoning guidance signals for LLMs.
>
> 2.  **Evidence Quality Assessment:** Both experimental settings utilize identical retrieval results with the same top-100 triples ranked by QR-GAT scores. However, LLMs perform significantly better when confidence scores are explicitly provided in the prompt format (head, relation, tail) (Confidence: 0.95). This suggests that LLMs leverage these numerical values to assess evidence reliability during the reasoning process.
>
> ---
>
> **Q2: Can you provide a dedicated ablation study focusing on the impact of confidence scores on LLM reasoning performance?**
>
> Beyond the core comparison, we explored how confidence-based filtering affects performance using GLM-4-Air:
>
> * **Confidence > 0.001:** Includes nearly all retrieved triples → Macro-F1: 35.63
> * **Confidence > 0.01:** More selective filtering → Macro-F1: 36.77
> * **Confidence > 0.1:** Aggressive filtering → Macro-F1: 34.40
>
> **Key Findings:**
> (a) **Confidence scores consistently enhance reasoning performance** across different models;
> (b) **Moderate confidence thresholds (>0.01) can further improve performance** by eliminating extremely low-confidence noise;
> (c) **Overly aggressive filtering (>0.1) degrades performance** by removing necessary reasoning context;
> (d) **Overly permissive filtering (>0.001) introduces excessive noise** that impairs reasoning.
>
> This ablation study definitively demonstrates that confidence scores provide measurable reasoning improvements beyond retrieval quality enhancement alone.
>
> ## Q3: Inference Efficiency Comparison
> **Q3: How does QSRAG directly compare with existing iterative methods in terms of inference efficiency (latency, computational cost)?**
>
> Thank you for highlighting this crucial practical consideration. We conducted comprehensive efficiency analysis that demonstrates significant computational advantages:
>
> 1.  **End-to-End Latency Comparison:** SubgraphRAG achieves the best end-to-end performance among all baseline methods, taking only 6s compared to 948s of RoG and 68s of GNN-RAG, becoming our most competitive efficiency baseline.
>
> 2.  **Retrieval Efficiency Analysis:** Our detailed timing breakdown reveals:
>
> | Method          | WebQSP Retrieval Time | CWQ Retrieval Time |
> |----------------|------------------------|---------------------|
> | QSRAG          | 0.1006s                | 0.0920s             |
> | SubgraphRAG    | 0.0115s                | 0.0127s             |
>
> While QSRAG's retrieval is approximately 8-9x slower than SubgraphRAG, both methods operate within comparable time scales (0.01s vs 0.1s), demonstrating minimal computational overhead.
>
> 3.  **Key Efficiency Advantages:**
>     * **Single-pass retrieval and reasoning:** Unlike iterative methods such as RoG, QSRAG performs evidence retrieval once followed by single-pass LLM inference.
>     * **Comparable LLM inference processes:** Since both QSRAG and SubgraphRAG employ similar workflows, time differences stem from retrieval rather than reasoning complexity.
>     * **Superior accuracy-efficiency trade-off:** Our modest retrieval overhead (additional 0.08-0.09s) yields substantial accuracy improvements (+2.5% Triplet Recall on WebQSP, +11.3% on CWQ) while maintaining competitive efficiency.
>
> This analysis confirms QSRAG provides a practically viable solution with good accuracy-efficiency balance.
>
> ## Q4: Multi-hop Reasoning Performance Analysis
> **Q4: What specific types of multi-hop reasoning questions show the most significant improvements, and where does QSRAG still struggle?**
>
> We conducted systematic analysis based on reasoning complexity measured by hop counts:
>
> | Method | Webqsp |  |  |  | CWQ |  |  |  |  |  |
> |:-----:|:-----:|:-----:|:-----:|:-----:|:-----:|:-----:|:-----:|:-----:|:-----:|:-----:|
> |  | Triplerecall |  | AnswerRecall |  | Triplerecall |  |  | Answerrecall |  |  |
> |  | 1(65.8%) | 2(34.2%) | 1(65.8%) | 2(34.2%) | 1(28.0%) | 2(65.9%) | >=3(6.1%) | 1(28.0%) | 2(65.9%) | >=3(6.1%) |
> | subgraph hRAG | 0.953 | 0.748 | **0.977** | 0.881 | 0.831 | 0.820 | 0.626 | 0.946 | 0.916 | 0.741 |
> | QSRAG | **0.955** | **0.809** | 0.970 | **0.888** | **0.945** | **0.902** | **0.726** | **0.969** | **0.968** | **0.749** |
>
> **Key Findings:**
> (1) **Strongest improvements on 2-hop reasoning** across both datasets, with particularly notable gains, suggesting our query-aware attention mechanism excels for moderately complex reasoning chains;
> (2) **Consistent multi-hop benefits on the complex CWQ dataset** across all hop counts;
> (3) **Current limitation:** Our analysis focuses on hop-count rather than semantic question types.
>
> **Future Work:** We recognize the need for fine-grained analysis based on semantic question categories (compositional, comparative, temporal reasoning) to provide deeper insights into QR-GAT's effectiveness patterns.
>
> We thank you again for these valuable insights that will help improve our work's clarity and impact.

---

> ### Author Response · Authors · 2025-08-06
>
> Dear Reviewer KWxb,
>
> I hope this message finds you well. As the discussion phase is approaching its conclusion, I wanted to kindly follow up to ensure that our responses have sufficiently addressed your concerns. If you have any further questions or comments, we would be very grateful to hear them. Your feedback is truly valuable to us, and we’re eager to make any necessary improvements based on your insights. Thank you again for your time and thoughtful review.

---

> ### Author Response · Authors · 2025-08-08
>
> Dear Reviewer KWxb,
>
> Thank you again for your thoughtful review of our paper.
>
> We are writing to respectfully follow up on our rebuttal. As the discussion phase is nearing its end, we wanted to check if our responses have successfully addressed your concerns. We would be very grateful for any additional comments or questions you might have at this time.
>
> Your feedback is invaluable to our work, and we are eager to make any necessary improvements based on your insights.
>
> Thank you for your time and consideration.
>
> Sincerely,
> The authors

---

> ### Author Response · Authors · 2025-08-09
>
> Dear Reviewer KWxb,
>
> We hope this message finds you well.
>
> As the author-reviewer discussion phase is ending in just a few hours, we are sending a final, respectful follow-up to our rebuttal. We would be grateful if you could take a moment to confirm whether our responses have addressed your concerns.
>
> Your final input is crucial for a fair and complete review process.
>
> Thank you once again for your valuable time and consideration.
>
> Sincerely,
> The authors

---

> > ### Comment · Area_Chair_d7io · 2025-08-09
> >
> > Dear Reviewer KWxb,
> >
> > Thank you once again for carefully reviewing the paper & providing valuable feedback.
> >
> > Could you please take a look at the author response and indicate whether they address your feedback and concerns?
> >
> > Best,
> > AC

---

### Note · Authors · 2025-08-12

Dear AC, SAC, and PC members,

We sincerely thank all reviewers for their valuable time and insightful feedback on our paper. We have carefully addressed all comments in our rebuttal and have been fortunate to engage in productive discussions with several reviewers.

We would like to express special thanks to Reviewer WXMu for their positive feedback on our clarifications and for raising their score. We will incorporate their excellent suggestions, including a more detailed discussion of our architectural design choices in the Method section and an integration of the "Rihanna" example into the Introduction to better motivate our work.

We also appreciate Reviewer m27H's constructive comments and their understanding of the scope of this rebuttal phase. Their feedback has been very helpful in shaping our future work.

Unfortunately, we have not yet received any follow-up from Reviewers VKgY and KWxb. While we understand that reviewers are busy, we respectfully sent a final follow-up email. We sincerely hope that our rebuttal has successfully addressed their initial concerns, and we kindly ask the AC to review our responses to their comments for a comprehensive assessment.

We believe our work, which presents an effective solution to a challenging problem in KG-RAG, has been significantly improved through this review process. We are fully committed to making a thorough revision based on the valuable feedback we have received.

Thank you very much for your time and for overseeing this process.

Sincerely, The authors

---

### Decision · Program_Chairs · 2025-09-17

**Decision:**

Reject

**Comment:**

This paper proposes a novel QSRAG framework, particularly the Query-Relational Graph Attention Network (QR-GAT) component, which directly injects query and relation semantics into the attention mechanisms helping LLMs answer complex questions using KGs. QSRAG achieves state-of-the-art performance on two standard KGQA benchmarks. Overall the paper is clearly written and easy to follow, has comprehensive evaluation and a novel component around KG-RAG. It is important to address all questions and concerns of the reviewers in the final version. Specifically strengthen the paper with the baseline comparisons; addressing the statistical rigor aspect as raised by the reviewer, more comparisons of the retrieval component and end-to-end time cost against the baseline.